# Comparative transcriptome profiles of *Schistosoma japonicum* larval stages: Implications for parasite biology and host invasion

**Shaoyun Cheng**[1☯], **Bingkuan Zhu**[1☯], **Fang Luo**[1], **Xiying Lin**[1], **Chengsong Sun**[2], **Yanmin You**[1], **Cun Yi**[1], **Bin Xu**[3], **Jipeng Wang**[1], **Yan Lu**[1], **Wei Hu**[1,3]*

**1** Department of infectious diseases, Huashan Hospital, State Key Laboratory of Genetic Engineering, Ministry of Education Key Laboratory for Biodiversity Science and Ecological Engineering, Ministry of Education Key Laboratory of Contemporary Anthropology, School of Life Science, Fudan University, Shanghai, China, **2** Anhui Provincial Institute of Parasitic Diseases, Hefei, China, **3** National Institute of Parasitic Diseases, Chinese Center for Disease Control and Prevention, Key Laboratory of Parasite and Vector Biology of China Ministry of Health, WHO Collaborating Centre for Tropical Diseases, Joint Research Laboratory of Genetics and Ecology on Parasite-host Interaction, Chinese Center for Disease Control and Prevention & Fudan University, Shanghai, China

☯ These authors contributed equally to this work.
* huw@fudan.edu.cn.

**Data Availability Statement:** All raw sequencing data are available via NCBI under SRA accessions PRJNA719283. Nucleotide sequences have been deposited in the Sequence Read Archive (SRA) of

## Abstract

*Schistosoma japonicum* is prevalent in Asia with a wide mammalian host range, which leads to highly harmful zoonotic parasitic diseases. Most previous transcriptomic studies have been performed on this parasite, but mainly focus on stages inside the mammalian host. Moreover, few larval transcriptomic data are available in public databases. Here we mapped the detailed transcriptome profiles of four *S. japonicum* larval stages including eggs, miracidia, sporocysts and cercariae, providing a comprehensive development picture outside of the mammalian host. By analyzing the stage-specific/enriched genes, we identified functional genes associated with the biological characteristic at each stage: e.g. we observed enrichment of genes necessary for DNA replication only in sporocysts, while those involved in proteolysis were upregulated in sporocysts and/or cercariae. This data indicated that miracidia might use leishmanolysin and neprilysin to penetrate the snail, while elastase (*Sj*CE2b) and leishmanolysin might contribute to the cercariae invasion. The expression profile of stem cell markers revealed potential germinal cell conversion during larval development. Additionally, our analysis indicated that tandem duplications had driven the expansion of the papain family in *S. japonicum*. Notably, all the duplicated cathepsin B-like proteases were highly expressed in cercariae. Utilizing our 3rd version of *S. japonicum* genome, we further characterized the alternative splicing profiles throughout these four stages. Taken together, the present study provides compressive gene expression profiles of *S. japonicum* larval stages and identifies a set of genes that might be involved in intermediate and definitive host invasion.

NCBI under accession codes SRR14133806-SRR14133817. The genome assembly and gene set of SjV3 are available through Zenodo (https://doi.org/10.5281/zenodo.5795038).

**Funding:** The project was foundeded by the National Key research and Development Project of China (Award Number: 2018YFA0507300) and the National Natural Science Foundation of China (Award Number: 31572513) to Wei Hu. The funders had no role in study design, data collection and analysis, decision to publish, or preparation of the manuscript.

**Competing interests:** The authors have declared that no competing interests exist.

## Author summary

Schistosomes are parasitic flatworms that require a snail host and a mammalian host to complete their life cycle. Due to the difficulties in obtaining materials, little is known about the molecular aspects of this fluke's larval stages. Based on RNA-Seq, we provide the first high-resolution, transcriptomic analysis of four larval stages of *Schistosoma japonicum*. The data showed the biological and physical features of each stage, also highlighted that miracidia and cercariae might use a different group of proteases for host invasion. Additionally, it indicated that different populations of germinal cells may exist in the larval stages. The high expression of tandem duplicated cathepsin B-like proteases at the cercariae stage may contribute to the wide definitive host range of *S. japonicum*. Additionally, we observed that alternative splicing plays a vital role in regulating gene expression in *S. japonicum*, among which skip exon was the most predominant. Our data provide valuable information on the expression and function of *S. japonicum* genes across their larval stages and will support basic and applied research for the community.

## Introduction

Schistosomiasis is a neglected tropical disease caused by *Schistosoma spp*., which is prevalent in approximately 78 countries and affects more than 240 million people worldwide [1]. Similar to other species, *Schistosoma japonicum* has a complex life cycle occurring in two hosts: snail and mammal. The life cycle outside the mammalian hosts involves eggs excreted with feces, asexually reproducing larvae in the snail host and two intermediate free-swimming stages. The mature egg is excreted with host feces and releases a miracidium in fresh water. Then the ciliated and free-swimming larva seeks the snail-*Oncomelania* genus guided by chemical attraction and further penetrates the snail host assisted by secretions, which are probably proteolytic enzymes [2]. Within the snail, the miracidium loses its ciliated plates and undergoes a dramatic developmental conversion into a mother sporocyst that contains a population of totipotent stem cells, called germinal cells. The germinal cell will proliferate and form the germinal balls (primordial daughter sporocyst embryos) to further produce daughter sporocysts. Upon maturation, the daughter sporocysts emerge from the mother sporocyst and migrate to the snail's hepatopancreas where they undergo development to produce the free-swimming cercariae by asexual reproduction [3]. Under suitable light and temperature conditions, the fork-tailed cercariae will shed from the snail. There are five pairs of acetabular glands inside the cercarial head which contain many kinds of proteases [4]. Once contact with the skin of human and mammals, the cercariae complete the penetration process by the mechanical activity of the tail and the hydrolytic activities of the proteolytic enzymes.

Transcriptomic studies across the life cycle of schistosomes have been extensively conducted, but mainly by ESTs, SAGE or microarray approaches [5–10]. These techniques lack the accuracy and sensitivity of a more contemporary RNA-seq approach, which is a powerful tool for delivering genome-wide transcription profiles unconstrained by genomic annotation. Previous studies applying RNA-seq mainly focused on adult stages, gonads [11–14] or parasites derived from different hosts [15,16]. Dynamic transcriptome profiles of *S. japonicum* and *S. mansoni* from juvenile schistosomulae to sex mature adult worms offered many insights into the reproductive development of the parasites during intra-mammalian development [17,18]. Recently, the intramolluscan transcriptomes of *S. mansoni* were investigated [19], but the samples used for RNA-Seq were infected whole snails. Thus, the landscape of gene

expression in ex vivo larval stages, especially the molecular basis behind the stage transitions, is still lacking.

To obtain the gene expression information in the larval stages, we performed RNA-seq analysis on *S. japonicum* eggs, miracidia, sporocysts and cercariae. We used the most recent 3[rd] version of *S. japonicum* genome for mapping and annotation. Gene expression information correlated well with the biology of each life stage. Miracidia and cercariae showed high motor and proteolysis activity, ready for the host invasion. DNA replication and cell division only occurred in the sporocysts. We identified genes that are stage specifically expressed or with enriched expression that could thus be vital for the dominant functions of the parasite in those life stages. We found that each larval stage has germinal cells and there may be germinal cell conversion during the larval development. Interestingly, we discovered the tandem duplication events drove the expansion of the papain gene family in *S. japonicum*. Furthermore, we identified a large number of alternative splicing (AS) events in each stage, indicating that AS is a widespread process for generating protein isoform diversity in *S. japonicum*. This study provides rich and valuable resources for the community to understand the larval biology and will assist in the exploration of novel anti-schistosome targets and vaccine candidates.

## Materials and methods

### Ethics statement

All experiments involving animals were carried out in accordance with the guidelines for the Care and Use of Laboratory Animals of the Ministry of Science and Technology of the People's Republic of China (2006398) and approved by the Ethics and Animal Welfare Committee of the National Institute of Parasitic Diseases, Chinese Center for Disease Control and Prevention, Shanghai, China (IPD2008-4).

### Parasite material

All parasite material was from an Anhui isolate of *S. japonicum* maintained in the National Institute of Parasitic Diseases, Chinese Center for Diseases Control and Prevention, Shanghai.

### Isolation of eggs

Eggs were isolated by an improved enzymatic method. Three New Zealand white rabbits were infected with 800–1000 cercariae each. Six weeks later, the liver tissues were chopped with a scalpel blade and homogenized in 500 mL saline solution. The suspension was successively passed through 80 and 180 mesh metal sieves. After repeated centrifugation and removing the tissue debris, the pellet was resuspended in 10 mL saline solution containing 100 μg collagenase IV (Solarbio Life Sciences, Beijing), then incubated at 37˚C for 30 min with gentle shaking. The sample was then centrifuged at 2,000 rpm at 4˚C for 8 min, and the residues after digestion were removed. The egg pellet was then washed twice with saline solution. Finally, the eggs were washed by pipetting on a 300-mesh nylon screen, then collected and stored in 1.2% NaCl solution at 4˚C under dark. Eggs isolated from one rabbit were used as a biological replicate.

### Isolation of the miracidia

Purified eggs were transferred into a 200 mL hatching measuring cylinder wrapped completely in light-blocking black tape with the exclusion of the top 4 cm from the lip, thereby producing a light gradient. The hatching cylinder was topped with artificial pond water (0.46 μM FeCl3 ·6 H2O, 220 μM CaCl2 ·2 H2O, 100 μM MgSO4 ·7 H2O, phosphate buffer [313 μM KH2PO4,

14 μM (NH4)2SO4] pH 7.2) [4] until above the tape-covered area ~1.5cm and exposed to bright light at 28°C. Eggs were incubated for 2 h post-hatch, and the top 10 mL of miracidia-containing water (MCW) was collected for miracidia isolation. Hatched miracidia were collected by centrifugation at 8000 × *g* for 1 min at 4°C, and were then washed twice with water. Miracidia hatched from the eggs purified from the liver of one rabbit were used as one replicate.

## Isolation of the daughter sporocysts

Daughter sporocysts were separated from the hepatopancreas of *Oncomelania hupensis* snails after ~5–6 weeks post-infection with dissecting needles under an optical microscope and purified after washing three times with sterilized PBS (pH 7.4). Daughter sporocysts collected from 6~10 infected snails were used as one replicate.

## Collection of cercariae

To obtain *S. japonicum* cercariae, *O. hupensis* snails ~7–8 weeks post-infection were exposed to light in artificial pond water at 26°C for 2 h. The emerging cercariae were gravity-concentrated by cooling on ice for two hours, which prevented swimming, then concentrated by centrifugation. Cercariae collected from 50~80 infected snails were used as one replicate.

## RNA isolation, library preparation and sequencing

Each larval stage has three biological replicates. For each replicate, we used 100~150 mg eggs, 15, 000~20, 000 miracidia, 1~2 mg daughter sporocysts and 15, 000~20, 000 cercariae. Parasites were homogenized in a 1 mL sterilized glass tissue grinder (Solarbio Life Sciences, Beijing) and total RNA was isolated using Qiagen RNeasy Micro Kit (Valencia, CA). RNA quality was assessed by 1% agarose gel electrophoresis and a NanoPhotometer spectrophotometer (Implen, Westlake Village, CA, USA). RNA integrity was assessed using the RNA Nano 6000 Assay Kit of the Bioanalyzer 2100 system (Agilent Technologies, CA, USA). RNA-Seq libraries were generated with the NEBNext Ultra Directional RNA Library Prep Kits (NEB, USA) according to the manufacturer's protocol. After clusters generation on a cBot Cluster Generation System using TruSeq PE Cluster Kit v3-cBot-HS (Illumia), the libraries were sequenced on an Illumina Novaseq platform (Novogene, Tianjin, China) with paired-end 150 bp.

## Read mapping and data processing

Quality control (QC) of the raw sequencing data was performed using the FASTQC program. Low-quality reads and adapter sequences were trimmed using fastp tool v.0.20.1. (parameters: -q 15 -u 40 -n 5 -l 15) [20]. The clean reads were mapped to the chromosome-level *S. japonicum* reference genome (*Sj*V3) using STAR 2.4.2a in twopassMode (parameters:—outFilterMultimapScoreRange 1—outFilterMultimapNmax 10—outFilterMismatchNmax 10—alignIntronMax 500000—sjdbScore 2—alignSJDBoverhangMin 3) [21] and further used to estimate the transcript abundance in TPM (Transcripts Per Kilobase million) using RSEM v.1.3.1 [22] with default parameters. These transcript abundances were imported into R and summarized with tximport v1.18.0 [23]. Principal Component Analysis (PCA) was performed using the prcomp function in the stats (v3.6.0) R package. Hierarchical clustering analysis (HCA) was performed with pheatmap (https://cran.r-project.org/web/packages/pheatmap/index.html). The R package DEseq2 v1.26.0 [24] was used to perform differential expression analysis. Gene Ontology (GO) enrichment analysis was performed with the R package

clusterProfiler [25]. The *P*-values were corrected for multiple hypothesis testing with the Benjamini–Hochberg false-discovery rate procedure (adjusted *P*-value).

## Gene clustering

To compare time-series gene expression data, the TPM values were clustered using R package Mfuzz v 2.46.0. Soft clustering was run with fuzzifier parameter set to m = 2.51, which was estimated using the mestimate() function. The cluster number was manually set at c = 8. Cluster members at a filter of 0.5 were used in subsequent enrichment analyses.

## Identification and phylogenetic analysis of papain gene family

The protein and genome sequences of *S. japonicum* (*Sj*V3) were downloaded from the Zenodo website (https://doi.org/10.5281/zenodo.5795038). The protein and genome sequences of *S. mansoni* were downloaded from the WormBase ParaSite (https://parasite.wormbase.org/index.html). The peptidase C1 (papain) family is part of clan CA of cysteine peptidases containing catalytic Cys25 (hereinafter papain numbering) and His159 residues in the active site [26]. To identify the papain genes in *S. japonicum* and *S. mansoni*, the HMMER profile of peptidase C1 domain (PF00112) was from the Pfam database and searched against the protein sequences of *S. japonicum* and *S. mansoni* with an *E* value cutoff of $1 \times 10^{-40}$. To study the phylogenetic relationships of the papain genes between *S. japonicum* and *S. mansoni*, multiple sequence alignments of amino acid sequences were performed using the ClustalW program with default parameters. An unrooted neighbor-joining tree was constructed with 500 bootstrap replications using MEGA 7.0 software based on the full-length protein sequence alignment. All identified papain genes were classified into different groups based on gene annotation and the alignment results. Consensus Newick format trees were compiled with MEGA 7.0 software and edited with Adobe Illustrator.

## Characterization of the gene structures, conserved motifs and chromosomal distributions of the papain genes

Gene annotations of the identified papain genes were extracted from the genome reference GFF files. The conserved domains were analyzed using MEME 5.1.1 software and the maximum number of motifs was set to 10. The gene structures and motif patterns were drawn using TBtools software [27]. Collinearity analysis of papain protein sequences from *S. japonicum* and *S. mansoni* were performed with *BLAST*P and MCScanX software [28]. Duplication of papain genes was analyzed using MCScanX with *E*-value $< 1 \times 10^{-5}$. The R package "pheatmap" was used to draw the heatmap of papain genes based on their expression levels across the four *S. japonicum* life stages.

## Alternative splicing analysis

Alternative splicing (AS) events in these four *S. japonicum* life stages were detected and quantified using rMATs-turbo [29] with the reference transcript annotation. Percent spliced in (PSI) values were calculated for five classes of alternative splicing events, including skip exons (SE), retained introns (RI), mutually exclusive exons (MXE), alternative 5' splice sites (A5SS), and alternative 3' splice sites (A3SS). In total, AS events with a false discovery rate (FDR) < 0.05 between two adjacent stages were identified as differential AS events. Sashimi plots used to show splicing events were generated by rmats2sashimiplot (https://github.com/Xinglab/rmats2sashimiplot).

## Results and discussion

### An overview of the transcriptomes

Twelve digital expression (DGE) libraries were constructed and sequenced using the total RNA isolated from four *S. japonicum* stages (eggs, miracidia, sporocysts and cercariae) with three biological replicates (Fig 1A). Using RNA-Seq, we obtained 40,499,690 to 57,349,268 raw reads and 40,383,584 to 57,153,380 clean reads for each library. 85.84% to 92.31% of the clean reads were mapped to the 3$^{rd}$ version of the *S. japonicum* reference genome (S1 Table). To study the relationship between samples and evaluate the reproducibility of the biological replicates, we performed Principal component analysis (PCA), Pearson's rank correlation analysis, and hierarchical clustering analysis (HCA). The PCA plot shows the top two principal components that explain most of the variance between samples in the data set, 70% and 17% for PC1 and PC2. Replicates from eggs and miracidia were clustered closely, and the sporocysts and cercariae were clustered away from other stages (Fig 1B). Pearson's rank correlation analysis confirmed high reproducibility and consistent quality among the biological replicates (S1 Fig). Based on HCA, the expression profiles of the four life stages are clearly separated (S2 Fig).

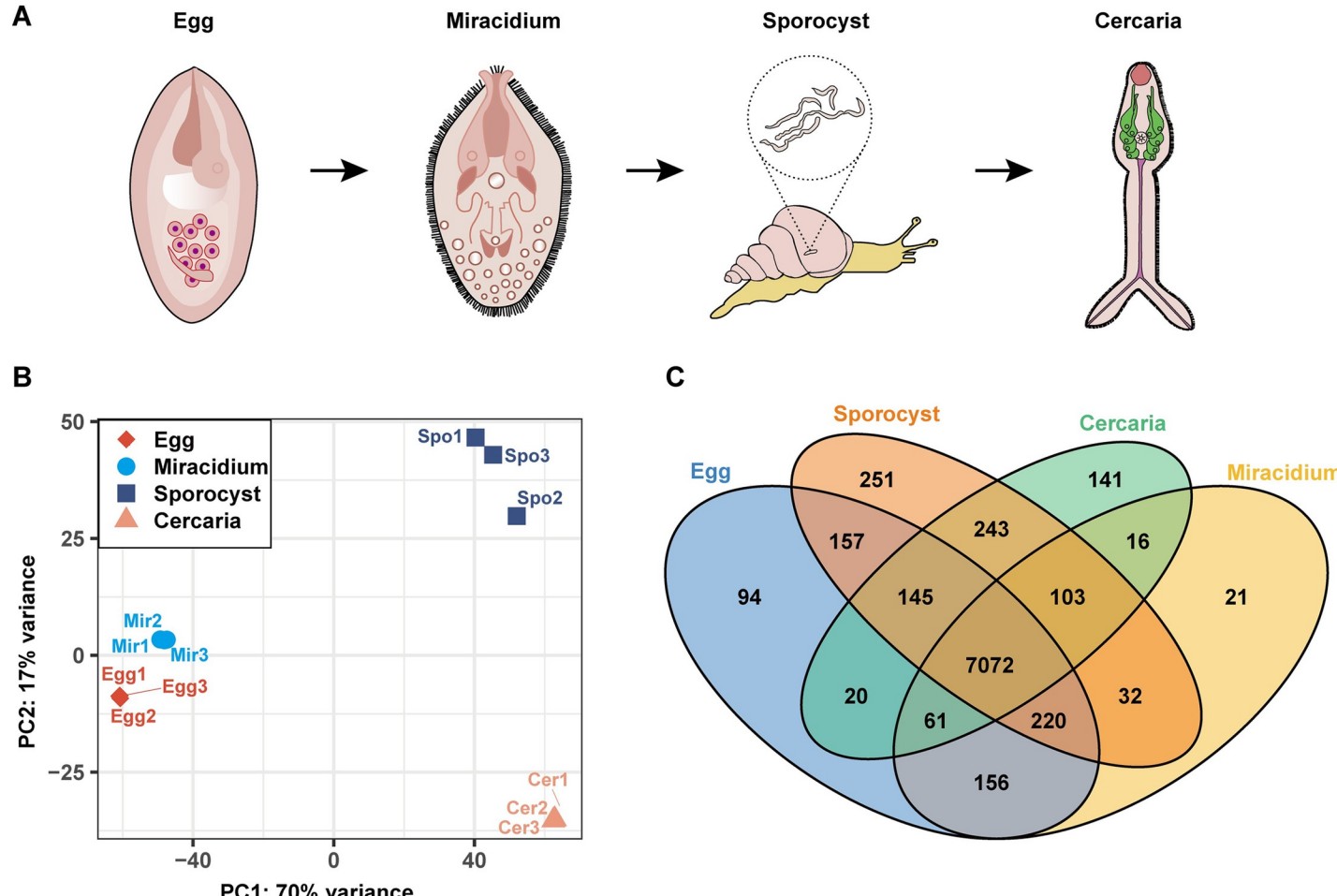

**Fig 1. Global transcriptomic profiles of *S. japonicum* in four different life stages.** RNA-Seq was performed on replicate samples of eggs (Egg), miracidia (Mir), sporocysts (Spo) and cercariae (Cer). (A) Four *S. japonicum* larval stages used for RNA-Seq. (B) PCA results. Each symbol indicates an individual sample. (C) Venn diagram showing differentially transcribed genes among the four life stages. Circles of different colors represent a set of genes transcribed in one stage, where the values represent the number of uniquely transcribed genes in one stage, or common transcribed genes between two, three or four life stages.

Additionally, these analyses revealed a high similarity between eggs and miracidia, which is reasonable because some eggs contain well-developed miracidia. To obtain the co-expressed and differentially expressed genes between the four stages, an inter-sample Venn diagram analysis was performed. The mRNA with an expression value equal to or greater than 1 TPM in at least two of the three biological replicates was considered expressed. A total of 8,732 genes were identified, and 7,925, 7,681, 8,223, and 7,801 genes were found in eggs, miracidia, sporocysts and cercariae, respectively. The number of genes identified in each larval stage was much higher than that reported by Gobert et al. [10] and Cai et al. [9]. Most of the genes (7,072) were identified in all four life stages, but 94, 21, 251, and 141 genes were exclusively detected in eggs, miracidia, sporocysts and cercariae (Fig 1C and S2 Table).

## Stage-specific genes (SSG) and stage-enriched genes (SEG)

Since the parasite at each stage shows distinct biological characteristics, we attempted to identify genes expressed specifically at each stage, or significantly more highly expressed at each stage. We thus defined SSG (genes expressed at only one stage) and SEG (genes have significantly higher expression at one stage compared to the other three with a FC > 5 and FDR < 0.05) to further analyze the genes related to their stage-related features. We obtained 94 SSG and 245 SEG at the egg stage, 21 SSG and 2 SEG in miracidia, 251 SSG and 255 SEG in sporocyst as well as 141 SSG and 209 SEG in cercaria (Fig 2A and S3 Table). By performing GO analysis, we explored the enriched functions of SSG and SEG at each life stage (Fig 2B and S4 Table).

## SSG and SEG in eggs

SEA (soluble egg antigen) secreted by egg induces granulomas in the mammalian host. At this stage, four members of T2 ribonucleases (*Sjc*_0002258, *Sjc*_0002263, *Sjc*_0002281, and

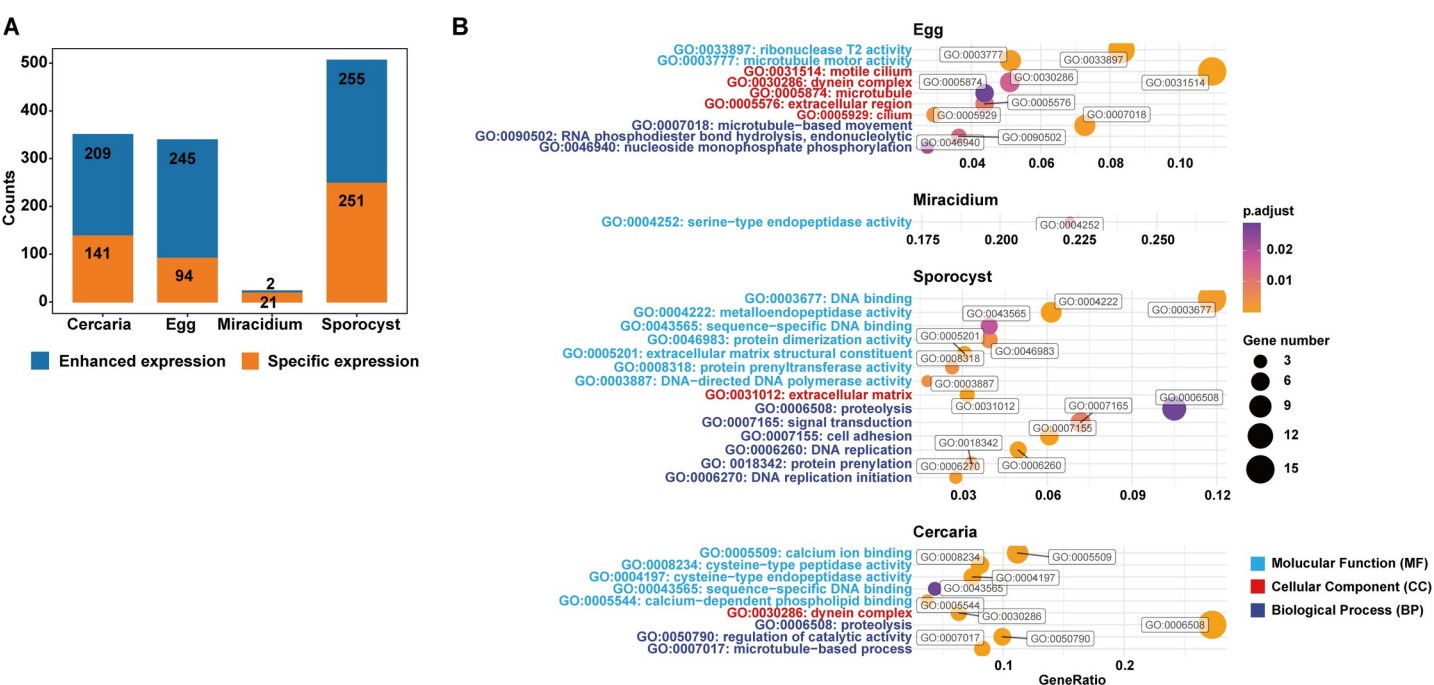

**Fig 2. Stage-specific genes (SSG) and stage-enriched genes (SEG) in the four larval stages.** (A) The number of SSG and SEG in each stage. (B) GO enrichment analysis based on the SSG and SEG in each stage. Gene ratio is the percentage of total SEG or SSG in the given GO term.

*Sjc*_0002296) associated with ribonucleases T2 activity (GO: 0033897) showed specific expression. T2 ribonuclease was identified as one of the top 25 most highly enriched genes in eggs in a previous study [9]. T2 ribonucleases are found in the genome of protozoans, plants, animals and viruses. A broad range of biological roles for these ribonucleases have been revealed, including scavenging of nucleic acids, degradation of self-RNA, serving as extra- or intracellular cytotoxins, and modulating host immune responses [30]. Previous study reported that *S. mansoni* omega-1, one major component of the SEA that was specifically expressed in eggs [31], was able to drive Th2 cell responses [32]. Omega-1 is a member of T2 ribonuclease family, both its RNase activity and glycosylation are essential for Th2 skewing [31]. Omega-1 can be taken up by dendritic cells and alter their cytoskeletal structure and function [33,34]. The protein structure analysis revealed that *S. mansoni* Omega-1 and the four *S. japonicum* T2 ribonucleases all contained ribonuclease_T2 domain (S3 Fig). Therefore, we propose that these T2 ribonucleases specifically expressed in eggs may participate in the regulation of host immune response.

## SSG and SEG in miracidia

Miracidia are free-swimming and penetrate snail host. We noticed that ten members of cilia- and flagella- associated protein (CFAP) (*Sjc*_0004811, *Sjc*_0003323, *Sjc*_0007156, *Sjc*_0000523, *Sjc*_0002745, *Sjc*_0000347, *Sjc*_0007012, *Sjc*_0000734, *Sjc*_0006213, and *Sjc*_0000078) related to motile cilium (GO: 0031514) were enriched at this stage. CFAP plays a role in the reproduction of the size and morphology of cilia [35]. Therefore, it may be involved in the assembly of the cilium during the development of miracidia. Alternatively, CFAP was reported to regulate olfactory transduction in mice [36]. Since miracidia are attracted by the chemical substances secreted by the snail [37] to locate the host, CFAP may participate in this process. In addition, a 5-hydroxytryptamine (5-HT, serotonin) receptor (*Sjc*_0004493) was only detected at this stage. The neurotransmitter molecule 5-HT regulates diverse physiological processes in both invertebrates and vertebrates [38]. In schistosomes, 5-HT treatment significantly stimulated motility of the *in vitro* cultured sporocysts or adult worms [39,40]. And its receptor *Sm*5HTR was required for the proper control of motility in *S. mansoni* [41]. Thus, we reasoned that this 5-HT receptor may be responsible for the control of movement in miracidia.

## SSG and SEG in sporocysts

Sporocysts are resident in snail hosts and produce cercariae. One cercarial elastase (*Sj*CE2b, *Sjc*_0008947) and nine leishmanolysins (*Sjc*_0006363, *Sjc*_0006650, *Sjc*_0006649, *Sjc*_0006646, *Sjc*_0006648, *Sjc*_0006647, *Sjc*_0006120, *Sjc*_0006219, and *Sjc*_0006399) associated with proteolysis (GO: 0006508) were specifically expressed at this stage. There was an expanded family of elastases in *S. mansoni* [42], but only one member in *S. japonicum* [43]. Cercarial elastases are the major invasive proteases in *S. mansoni* [44], and are considered to be involved in *S. japonicum* cercariae invasion as well [45]. Leishmanolysin (also called GP63) is a critical virulence factor in various *Leishmania* species. This important metalloprotease manipulates the host immune system by allowing the parasite to establish, survive and propagate within mammalian macrophages [46]. Recently, it was reported that a leishmanolysin derived from *S. mansoni* excretory/secretory products could interfere snail haemocyte morphology and migration [47]. It's worth noting that the cercaria head is mostly transcriptionally and translationally quiescent [48]. The elastase and leishmanolysins transcribed in the daughter sporocysts may be prepared for the cercariae penetration. Additionally, genes necessary for DNA replication (GO: 0006160) and DNA replication initiation (GO: 0006270) were enriched in daughter sporocysts, such as DNA replication licensing factor MCM2 (*MCM2*, *Sjc*_0000705), G2/mitotic-specific

cyclin-B2 (ccnb2, *Sjc*_0005264) and mitotic spindle assembly checkpoint protein MAD2A (*MAD2L1*, *Sjc*_0002379). These genes play pivotal roles in the mitosis process, which is consistent with active cell division to produce cercariae at this stage. We also noted that more hypothetical proteins were enriched in daughter sporocyst than those from other stages, which may contribute to the biological functions of this poorly studied stage.

## SSG and SEG in cercariae

Cercariae are free-swimming and invade the mammalian host. One rhodopsin-type GPCR (*Sjc*_0002522) and two octopamine receptors (*Sjc*_0007275, *Sjc*_0000926) were specifically transcribed at this stage. Rhodopsin-type GPCRs were known to be involved in photoreception, typically thought of as light sensors in animals [49]. Given that the release of cercariae from snails is triggered by the sunlight, we speculated that it may participate in cercariae photokinesis. Octopamine (OA) is one of the invertebrate-specific biogenic amines. In locusts and mollusks, OA is involved in motor control [50]. In *S. mansoni* adult worms, OA is widely distributed in neurons of the peripheral nerve net that innervate muscle [51]. The octopamine receptors that specifically expressed in cercariae may play a role in movement control. Besides, calcium binding protein (CaBP, *Sjc*_0006317) related to calcium ion binding (GO: 0005509) exhibited the most enriched expression in the cercariae stage. The divalent cation calcium is used as a cellular signal or ionic cofactor involved in diverse metabolic processes, including secretion, metabolism, muscle movement and neuronal function [52]. In *Paragonimus ohirai*, the excystment of metacercaria is a calcium-dependent process [53]. In *Trichobilharzia regenti*, CaBP represented the fifth most differentially transcribed gene between cercariae and schistosomulae [54]. In *S. mansoni*, the preacetabular glands of cercariae contain a high concentration of calcium [55], and there is a downregulation of CaBP in cercariae following epidermal penetration [56]. These findings suggest that the function of calcium may be conserved among trematode species by regulating the larval physiology, while CaBP may be important for the infectious cercariae.

## Differentially expressed genes (DEG) between adjacent life stages

During development, *S. japonicum* undergoes dramatic morphological changes as well as physiological changes. To view the transcriptional changes during the transition of these four stages, pairwise differential gene expression analysis was performed for adjacent life stages. A list of differentially expressed genes (DEG) with a *P*-value < 0.05 was generated for each of the different pairwise comparisons of the life stages. A total of 1,184 DEGs were associated with the development from egg to miracidium, of which 190 and 994 were up-and downregulated in miracidium, respectively. 3,530 DEG between miracidium and sporocyst while 3,066 DEG between sporocyst and cercaria were detected (Fig 3 and S5 Table). Besides, Gene Ontology (GO) enrichment analysis was performed based on the DEG to analyze the enriched gene functions, the fold change cutoff value was set at > 5 to obtain more representative GO terms (S4 Fig and S6 Table).

During the development from egg to miracidium, two GO categories were upregulated, including magnesium ion binding (GO: 000287). The associated gene was a 7-methylguanosine phosphate-specific 5'-nucleotidase (*Sjc*_0000850). 5'-nucleotidases are enzymes catalyzing the hydrolytic dephosphorylation of nucleoside monophosphates to nucleosides and orthophosphate [57]. As catabolic enzymes, they play important roles in the regulation of nucleotide levels in living cells [58]. Compared to miracidia, fifteen GO categories were upregulated in eggs, including those related to lipid metabolic process (GO: 0006629), pyridoxal phosphate binding (GO: 0030170) and heme binding (GO: 0020037). Schistosomes are reported to

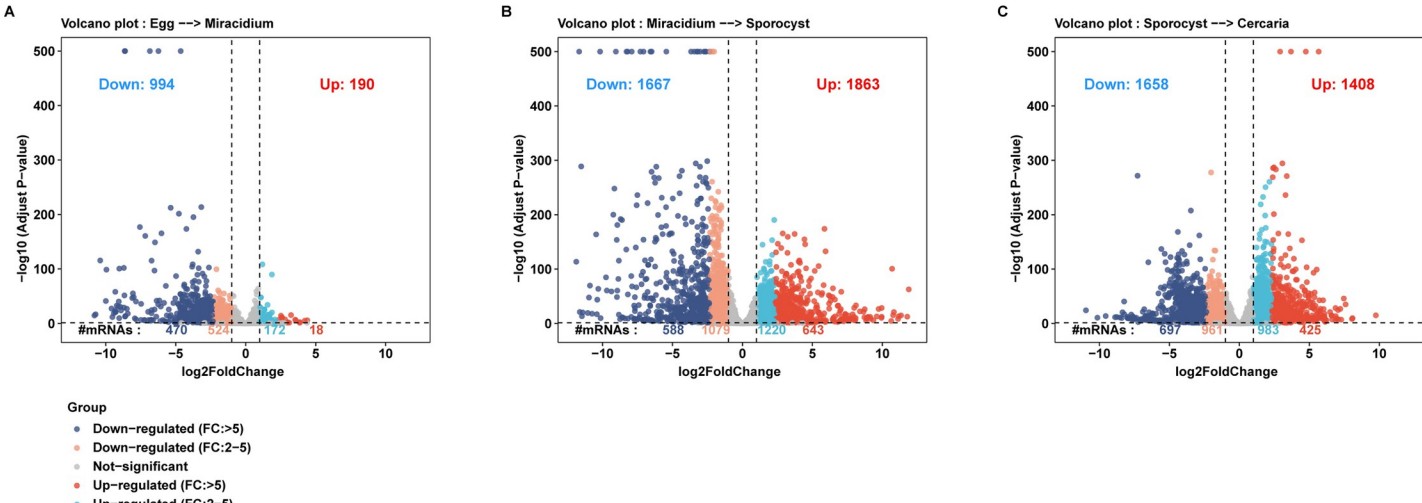

**Fig 3. Differential gene expression among the stage transitions.** Volcano plots showing differentially expressed genes (DEGs) in miracidium compared to egg (A), in sporocyst compared to miracidium (B), and in cercaria compared to sporocyst (C). Genes with *p*-value near 0 were adjusted to the *p*-value = $10^{-500}$. The dot line on the y axis refers to *p*-value = 0.05. The dotted lines on the x axis refer to fold change = -2 or 2.

uptake the basic sterols and fatty acids from host [59]. The pyridoxal phosphate (the active form of vitamin B6) [60] as well as heme, the important cofactor for oxygen transfer [61] were also derived from the host. Since the egg is produced and developed in the host, the high lipid metabolism activity and binding activity of vitamin B6 and heme may contribute to nutrients and oxygen that are required for the embryonic development in the egg.

During the development from miracidium to sporocyst, ten GO categories were enriched from the upregulated genes in miracidia. The enrichment of protein glycosylation (GO: 0006486) and fructosyltransferase activity (GO: 0008417) may be associated with the enhanced formation of glycocalyx that covers the surface of the miracidium [62]. Besides, a group of leishmanolysin and neprilysin linked to metalloendopeptidase activity (GO: 0004222) were upregulated. Notably, these leishmanolysins were different from those specifically expressed in sporocysts. Neprilysin (NEP) is a zinc-metalloenzyme belonging to the M13 family [63], which plays an important role in the interactions between host and parasite. Neplilysin was shown to involve in the production of immunoactive peptides, which could inactive the immunocytes from the snail host [64]. These data indicate that leishmanolysin and neprilysin may help the parasite escaping the immune attack of the *Oncomelania hupensis* hemocytes after miracidia penetration. On the opposite, ten GO categories were upregulated in sporocysts, including DNA replication (GO: 0006260), DNA replication initiation (GO: 0006270) and DNA binding (GO: 0003677), which are consistent with the asexual reproduction of sporocysts that undergoes active cell division [19].

Ten GO categories were upregulated during the development from sporocyst to cercaria. Go terms related to glycolytic process (GO: 0006096), transmembrane transport (GO: 0055085) and microtubule-based process (GO: 0007017) represent the high motility and metabolic activity of cercaria. Besides, proteolysis (GO: 0006508) related cysteine-type (GO: 0004197), aspartic-type (GO: 0004190) and serine-type endopeptidase activity (GO: 0004252) were also enriched. Many members of these important protease family have been studied, such as cathepsin B and cathepsin L from the cysteine protease family [65], cathepsin D from the aspartic protease family [66] and trypsin and elastase from the serine protease family [67].

These proteases participate in numerous biological processes, such as the parasite's invasion, survival and longevity in the definitive host.

## Gene clusters based on expression pattern across the four life stages

To further explore the transcriptomic changes across the four life stages, genes were clustered into 8 groups based on their expression profile (Fig 4). Each cluster showed a specific expression pattern. Genes from clusters 1, 2 and 5 showed highest expression in eggs, miracidia and cercariae respectively, while those from clusters 3 and 4 were highly expressed in sporocysts. GO categories enriched in these clusters were generally consistent with those based on SSG and SEG of each stage (S7 Table). We noticed that cluster 5 contained 17 genes that are linked to G protein-coupled receptor (GPCR) signaling pathway (GO: 0007186), including one metabotropic glutamate receptor (*Sjc_*0001309), one tachykinin-like peptides receptor (*Sjc_*0003380), one rhodopsin GPCR (*Sjc_*0003938), two 5-HT receptors (*Sjc_*0001525, *Sjc_*0007274) and three octopamine receptors (*Sjc_*0005446, *Sjc_*0007275, and *Sjc_*0006297). Glutamate is a neurotransmitter that is involved in controlling parasite locomotion [68]. Tachykinins are peptides that play a role in the processing of sensory information and control of motor activities [69]. These GPCRs may coordinate to regulate cercariae movement, light response, host sensory and adaption to a new osmotic pressure environment after host penetration. Genes in cluster 6, 7 and 8 showed high expression at two stages. Among them, cluster 6 showed increased expression in both the miracidia and cercariae. The enriched GO categories were mainly related to protein synthesis and transport, such as endoplasmic reticulum (GO: 0005783), intracellular protein transport (GO:0006886) and vesicular mediated transport (GO: 0016192). This may reflect that parasites at these two free-swimming stages are preparing proteins used for the host invasion.

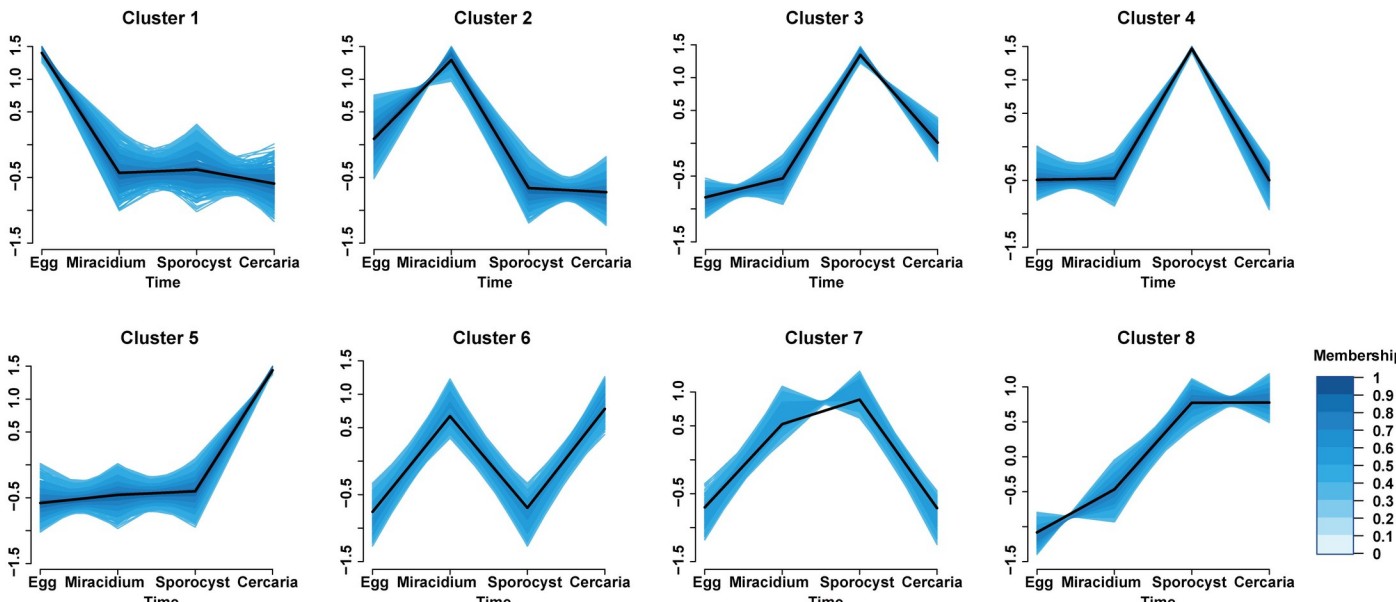

**Fig 4. Clusters of genes based on time-course expression pattern.** Membership value was calculated using R package Mfuzz, which indicated the degree of membership of this gene for the corresponding cluster. The y-axes were scaled independently to emphasize the differences between clusters.

## Genes associated with germinal cell and asexual reproduction in the larval stages

In planarians (close relatives of schistosomes), a population of pluripotent stem cells called neoblasts can regenerate injured tissues and replenish a whole animal from a single cell [70,71]. In recent years, it was shown that schistosomes, like the planarians, also contain stem cells. There are two major stem cells that play in the different life stages during intramolluscan and intramammalian development, including the germinal cells of sporocysts involved in asexual reproduction and the neoblast of adult worms involved in sexual reproduction [72]. Germinal cells have similar morphology with the neoblasts of planarians, they have a high nuclear to cytoplasmic ratio, an open chromatin structure and a large nucleolus [73]. These cells seem to proliferate indefinitely, evidenced by the serial transplantation of sporocysts into naïve molluscan hosts that led to continuous propagation of the parasites [74].

Recently, many methods were developed to study schistosome stem cells, including 5-ethynyl-2′-deoxyuridine (EdU) labeling, whole-mount in situ hybridization (WISH), and RNA interference (RNAi) [75]. Wang et al. compared gene expression profiles of miracidia and sporocysts 48 h post-transformation with the transcripts enriched in FACS-purified planarian neoblasts [76]. The same group also transcriptionally profiled stem cells from *Schistosoma mansoni* in vitro transformed mother sporocysts at single-cell levels. Three major stem cells classes were identified upon their respective markers: k-cells (that transcribe *klf* and *nanos-2*); φ-cells (that transcribe *fgfrA* and *fgfrB*); and δ-cells that produce transcripts of *nanos-2* and *fgfrA, B* [77]. ScRNAseq analysis was also performed in the 2-day old schistosomulae to better understand the cell types and tissue differentiation [78]. These works provided many valuable markers of schistosome stem cells. We picked 18 markers and identified the homologs in *S. japonicum* using reciprocal BLAST comparisons (*E*-value $< e^{-10}$) (S8 Table). Then we described the expression profiles of these marker genes in the four larval stages (Fig 5A). *Nanos-2* expression was observed in the eggs and miracidia, but declined in the sporocysts. The RNA-binding protein *nanos-2* has been showed to function in schistosomes as a conserved regulator of adult stem cells [79]. It has been proposed that there are two germinal cell subpopulations (*nanos-2*$^+$ and *nanos-2*$^-$ cells) in the mother sporocysts, with the later subpopulation proliferating more rapidly [76]. The two fibroblast growth factor receptors (*fgfr*)—*fgfrA* and *fgfrB*, showed different expression patterns across these stages. *FgfrA* is essential for maintenance for adult stem cells [79], it's expression in miracidia, sporocysts and cercariae was stable (Fig 5B). However, *fgfrB* showed relatively high expression in the sporocysts (Fig 5B), suggestive of its important role in germinal cells. A polo-like kinase (*plk*) gene highly expressed in sporocysts, polo-like kinases are important regulators of cell cycle progression and mitosis [80]. Cell cycle related transcripts (*h2a*, *cyclinB*, and *PCNA*) all showed relatively high expression in sporocysts (Fig 5B). *Ago2-1*, a stem cell marker and enriched in germinal cells [76], was also upregulated in sporocysts (Fig 5B). *ZFP-1* is a zinc finger protein which is essential for the specification of new tegument cells [81]. This gene was highly expressed at the cercariae stage (Fig 5B). Soria et al. discovered a novel stem/germ cell marker-calmodulin (*cam*) in the schistosomulae [78]. Calmodulin is a $Ca^{2+}$ transporter and is critical for miracidium-to-sporocyst transformation [82], for sporocyst growth and for egg hatching [83]. We found this gene was upregulated in the cercariae stage (Fig 5B). In conclusion, these stem cell marker genes displayed differential expression patterns across the four larval stages. This indicated that germinal cells existed in every larval stage and stem cell conversion may have occurred during the larval development.

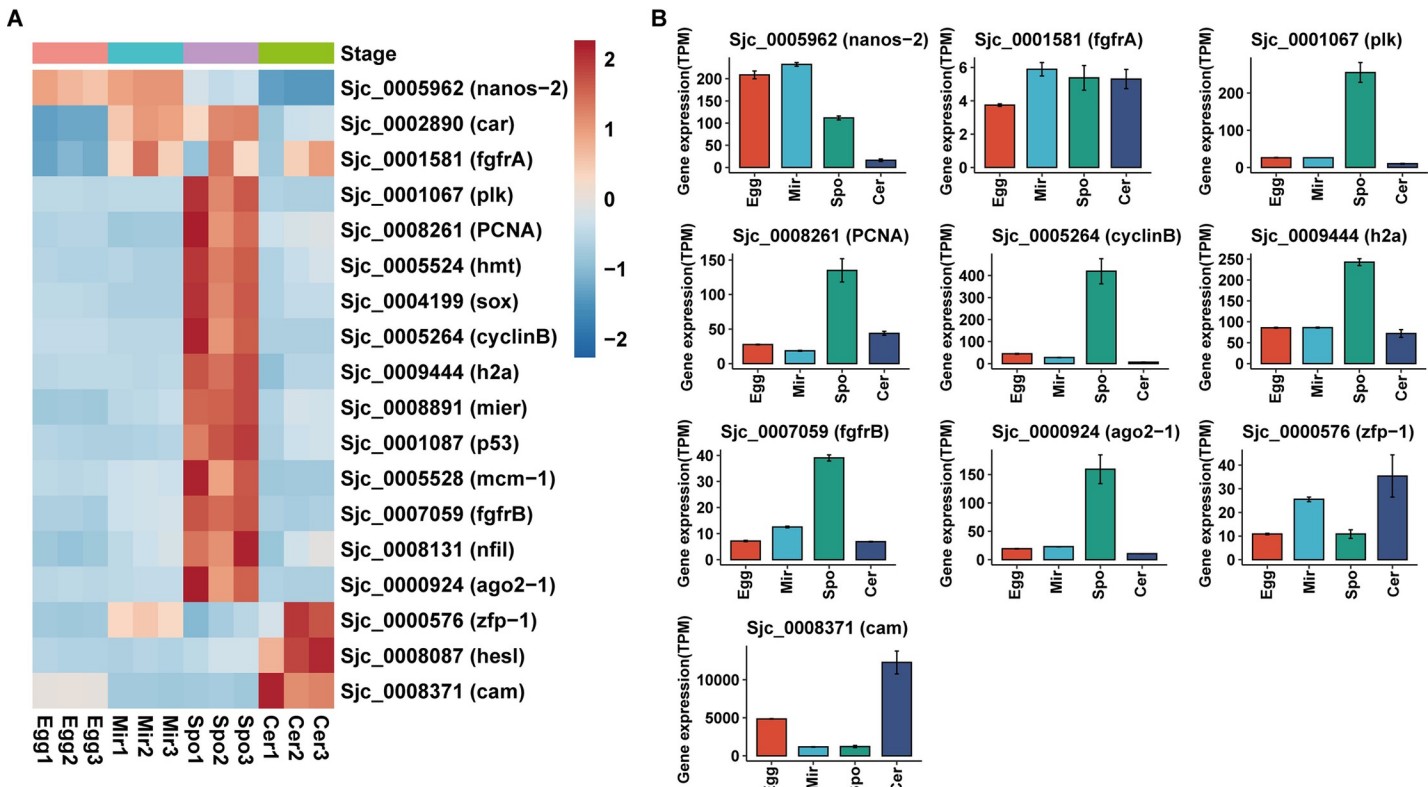

**Fig 5. Stem cell-related genes.** (A) Heatmaps showing the relative gene expression of the eighteen stem cell markers in egg, miracidium, sporocyst and cercaria, as indicated. (B) Expression profiles of the nine stem cell markers in the four larval stages.

## Highly expressed cathepsin-B like proteases in cercariae were produced by tandem duplications

Peptidase C1 genes are cysteine proteases of the papain superfamily. These enzymes play key roles in the pathogenesis of both protozoan and metazoan parasites, including skin or tissue penetration, hydrolysis of host or parasite proteins, and evasion or modulation of the host immune response [65,84]. The multiple functions of cysteine proteases make them attractive chemotherapeutic and vaccine targets [85].

Through a Hidden Markov Model (KMM) search, we identified 14 and 30 papain genes in *S. japonicum* and *S. mansoni* (S1 Dataset), respectively. To obtain the phylogenetic relationship of *Schistosoma* papain proteins, an unrooted tree was constructed. Papains were classified into six groups, including cathepsin B, cathepsin B-like, cathepsin C, cathepsin L, cathepsin L-like and cathepsin S (Fig 6A). A schematic representing the structure of all papain proteins was constructed from the MEME motif analysis results. As exhibited in Fig 6B, papain members within the same groups were usually found to share a similar motif composition. The exon-intron organizations of all the identified papain genes were examined to gain more insight into the evolution of the papain family. As showed in Fig 6B, all papain genes contain two to eleven exons (1with two exons, 5 with three exons, 27 with four exons, 1 with five exons, 5 with seven exons, 2 with eight exons, 2 with nine exons, and 1 with eleven exons). The papain genes in the same group shared similar gene structures.

To explore the relationship between *S. japonicum* and *S. mansoni* papain genes, we determined their chromosomal locations and whether they originated from gene duplication

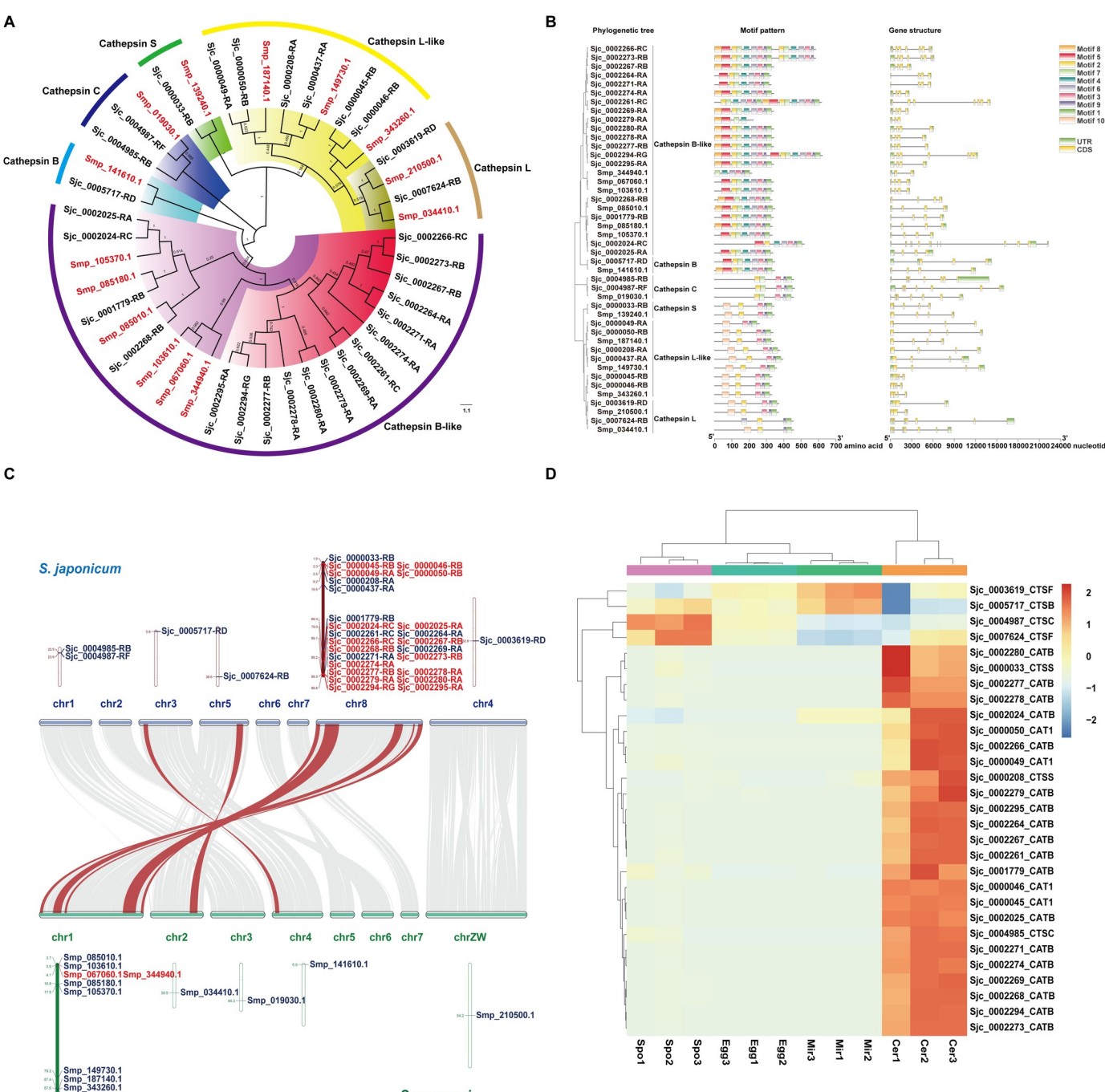

**Fig 6. Genome-wide identification, characterization, and expression patterns analysis of the papain gene family in *S. japonicum*.** (A) Phylogenetic relationship of papain. An un-rooted phylogenic tree was constructed in MEGA 7 using multiple alignments of *S. japonicum* and *S. mansoni* whole sequences data. (B) The motif composition and exon-intron structure of papain genes. For the motif pattern, the motifs, numbers 1–10, are displayed in different colored boxes. For the gene structure, green boxes indicate untranslated 5'- and 3'-regions; yellow boxes indicate exons; black lines indicate introns. (C) Synteny analysis of papain genes between *S. japonicum* and *S. mansoni*. Gray lines in the background show the collinear blocks within *S. japonicum* and *S. mansoni* genomes, while the red lines highlight the syntenic papain gene pairs. The tandem duplicated genes are marked in red. (D) Expression profiles of the papain genes at the four life stages.

events. We identified papain genes on five chromes, most of them are located on chromosome 8 (for *S. japonicum*) or chromosome 1 (for *S. mansoni*). Tandem-duplicated genes are defined as two paralogous genes that are separated by fewer than 10 intervening genes [86], and it is

one of the main sources of diversity for the evolution of gene families in eukaryotic organisms [87]. We identified 17 *S. japonicum* papain genes (10 pairs) that correspond to the tandem duplication events. In *S. mansoni*, we only detected 2 tandem-duplicated papain genes, comprising 1 gene pair (Fig 6C). We also added papain genes of *Schistosoma haematobium* and *Schistosoma bovis* into phylogenetic analysis. There are 14 papain genes in *S. haematobium* and 15 in *S. bovis*. The unrooted tree showed that a branch of *S. japonicum* cathepsin B-like proteases (14 members) separated from others, ten of them were produced by tandem duplication (S5 Fig). These results suggested that tandem duplication played important roles in the expansion of the papain gene family in *S. japonicum*. We then evaluated the papain gene expression profiles in different *S. japonicum* life stages via the RNA-seq data, and found that 87.5% of papain genes, including all the tandem-duplicated cathepsin B-like proteases, were highly expressed in the cercaria stage (Fig 6D).

Here, for the first time, we characterized the phylogenetic relationships, gene and protein structures and chromosome locations of the papain gene family in *S. japonicum*. We found that tandem duplication events drove the expansion of the papain gene family in *S. japonicum*. Besides, all the duplicated cathepsin B-like proteases showed the highest expression at the cercariae stage. *S. mansoni* and *S. haematobium* primarily infect humans, but *S. japonicum* is a zoonotic specie that infects more than 40 different mammalian species [88]. Thus, we speculate that these tandem duplicated and cercariae highly-expressed cathepsins may play important roles in assisting *S. japonicum* in establishing infections in broad definitive hosts.

## Genome-wide identifications and comparative analysis of alternative splicing events within the four life stages

Alternative splicing (AS) is the process that enables one gene to encode two or more mature mRNAs through the differential utilization of splice sites [89]. AS greatly expands the transcriptome and proteome diversity; it is widespread in the genomes of humans and other species [90,91]. In parasites, AS may play fundamental roles in the host-parasite interactions by producing alternative isoforms with different functions [92]. The discovery of AS giving rise to different isoforms of antigenic proteins is indicative of immune evasion strategies by the parasites [93]. AS has been investigated in *S. japonicum* schistosomulae and adult worms [94,95], but they were based on the early RNA-Seq technologies, the sequencing depth and accuracy were far inferior to the one used in our study. Here, based on the most accurate *Sj*V3 genome, we characterized the detailed AS in the four larval stages of *S. japonicum*.

In this study, five major AS types were considered as described before [96], including skip exons (SE), retained introns (RI), mutually exclusive exons (MXE), alternative 5' splice sites (A5SS) and alternative 3' splice sites (A3SS). 6,099, 6,101, 5,949 and 6,282 AS genes were determined in eggs, miracidia, sporocysts and cercariae (S2 Dataset), accounting for 62%, 62%, 61%, and 64% expressed genes, respectively (Fig 7A). The distributions of AS types in eggs, miracidia, sporocysts and cercariae are comparable; that is, SE was the most abundant AS events (44%-49%), followed by RI (21%-27%), A5SS (10%-12%), A3SS (10%-11%) and MXE (6%-9%) (Fig 7B).

The overall statistics of shared/unique AS genes in the four life stages are shown in Fig 7C. Interestingly, 5,374 genes were undergone alternative splicing in all four stages, and there are 143, 117, 124, 241 genes uniquely spliced at the stage of eggs, miracidia, sporocysts and cercariae, respectively. Notably, different AS patterns may occur for a single gene. Thus, UpSet plots were used to depict the intersections between AS types for each stage (Figs 7D and S6). For example, at the cercariae stage, a total of 2,240 genes only occurred in one type of AS event.

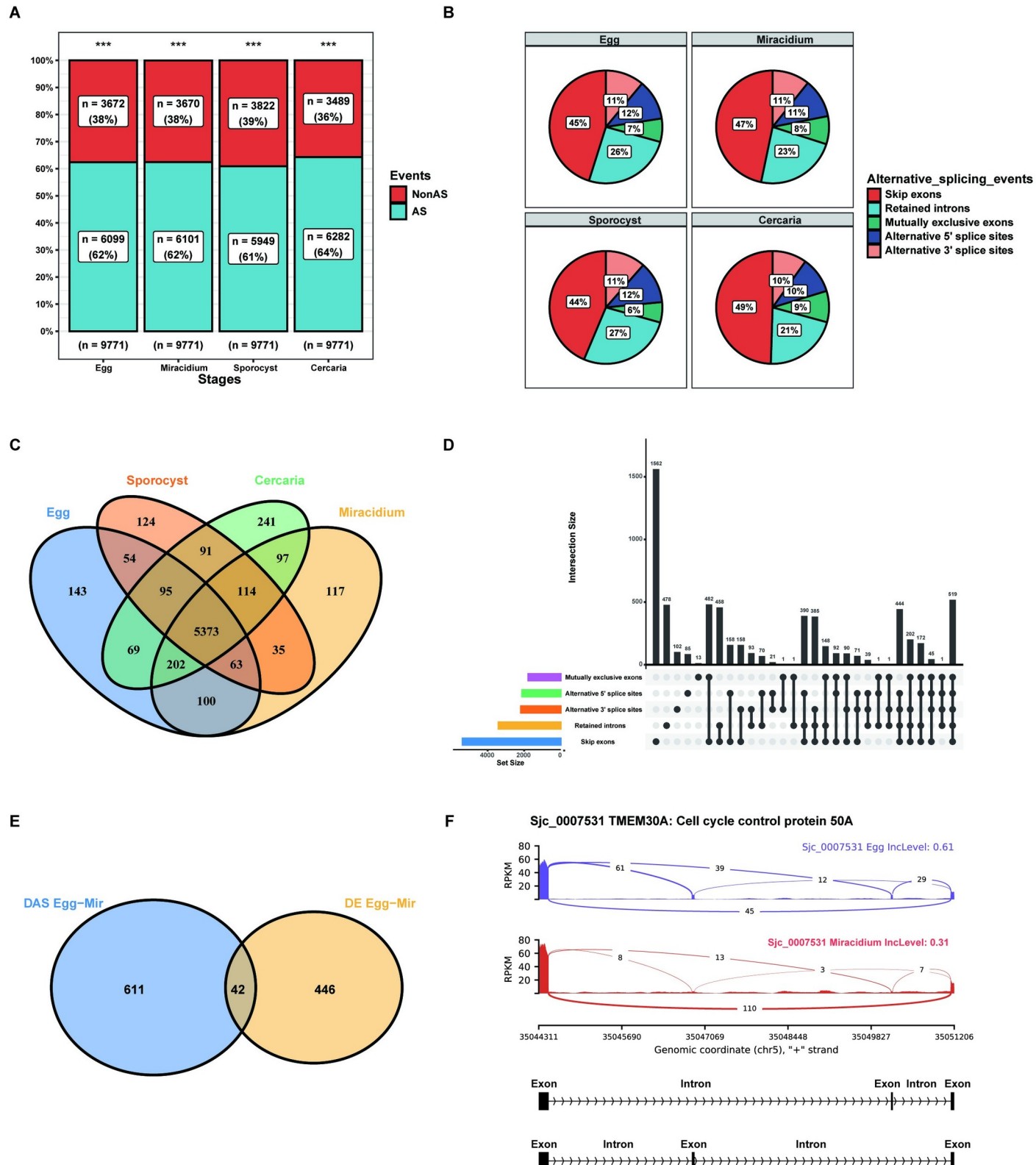

**Fig 7. AS landscapes in *S. japonicum* larval stages.** (A) Number and percentage of AS events in *S. japonicum* eggs, miracidia, sporocysts and cercariae. (B) Proportions of different AS types detected in *S. japonicum* eggs, miracidia, sporocysts and cercariae. (C) The Venn diagram shows the overlap of AS genes in the four life stages. (D) Interactions between the five types of detected AS genes in the cercaria stage were visualized using an UpSet plot. (E) Venn diagram of the overlap of the DE and DAS genes between the eggs and miracidia stages. (F) An example of a gene (*Sjc*_0007531) displayed different SE at the eggs and miracidia stages.

The number of the SE is the most, with 1,562 genes, and the least is MXE, with only 13 genes. Furthermore, 519 genes contained up to five types of AS events (Fig 7D).

To explore whether gene expression and AS acted cooperatively or independently to regulate *S. japonicum* development, both gene expression levels and gene with AS in the three stage transitions were simultaneously analyzed (S2 Dataset). Among the genes identified to be differentially expressed, only a small number overlapped with the genes had differential AS in the three comparisons: 42 (8.6%) between eggs and miracidia, 142 (11.5%) between miracidia and sporocysts, 106 (9.4%) between sporocysts and cercariae (Figs 7E and S7). *Sjc*_0007531, a cell cycle control protein, showed different SE in the eggs and miracidia (Fig 7F).

In conclusion, more than 60% of the expressed genes showed alternative splicing in each stage, much higher than previously detected in schistosomulae, which was 42.14%. The most common type of AS events detected in the larval stages was the SE, while the MXE was the least. In the schistosomulae and adult worms, the most common were SE and A3SS, the least were RI and MXE, respectively [94,95]. Among all events, RI is the predominant mode of AS in plants [97], whereas ES is the major type in humans [98]. However, in the cestodes *Echinococcus granulosus* and *Echinococcus multilocularis* or the free-living flatworm *Schmidtea mediterranea*, the major type AS was RI [99,100]. Hence, our results suggest that the gene regulation pattern of *S. japonicum* is much closer to its human hosts. Besides, we found that, similar to *Trypanosoma cruzi* and *Arabidopsis* [101,102], the co-regulated genes account only for a relatively small portion of all DAS or DE genes, which indicated that AS and gene activation could be separately regulated.

## Conclusions

This work presents the most thorough examination to date of the transcriptomes of *S. japonicum* larval stages. Evidence of DNA replication and cell division was only seen and confirmed in the sporocysts, while each stage upregulated different genes involved in development, morphogenesis, movement and host invasion. Our data indicated that neprilysins and leishmanolysins might play a role in the penetration of the snail by the miracidia. It's known that *S. japonicum* cathepsin B2 (*Sj*CB2) played fundamental roles in skin penetration [103]. Our analysis indicated that cercarial elastase (*Sj*CE2b) and leishmanolysins might also be involved in the process of cercariae invasion and the tandem duplications of cathepsin B-like proteases probably contributed to the wide mammalian host range of the *S. japonicum*. These genes should be targeted in the future for hypothesis-driven functional studies. The expression profile of stem cell markers revealed that different populations of germinal cells exist in the larval stages. We also performed the most comprehensive AS analysis in *S. japonicum*. We found that the AS prevalence was 61–64% at the genome-wide level, and ES was the predominant AS type in the larval transcriptomes, which revealed an affinity with its mammalian hosts in gene regulation patterns. The transcriptome profiles of *S. japonicum* larval stages provide new insights into host invasion, and the landscape of AS will not only facilitate future investigations on transcriptome complexity and AS regulation during the life cycle of *Schistosoma* species but also offered an invaluable resource for future functional and evolutionary studies of AS in platyhelminth parasites.

## Supporting information

**S1 Fig. A heatmap showing Person's correlation coefficient among different samples, with correlation levels indicated by colors.** The scores calculated by the R function () indicated the correlation levels between two samples.
(TIF)

**S2 Fig. Hierarchical clustering analysis (HCA) of transcriptional profiles from 12 *S. japonicum* samples with 8,732 genes.** Egg, egg; Mir, miracidium; Spo, sporocyst; Cer, cercaria.
(TIF)

**S3 Fig. Domain organizations of the *S. mansoni* Omega-1 and the four *S. japonicum* T2 ribonucleases.** The signal peptide and ribonuclease_T2 domain are depicted in red and blue, respectively.
(TIF)

**S4 Fig.** GO enrichment for differentially expressed genes (DEGs) in miracidium compared to egg (A), in sporocyst compared to miracidium (B), and in cercaria compared to sporocyst (C).
(TIF)

**S5 Fig. Phylogenetic relationship of papain in the four *Schistosoma* species.** The protein and genome sequences of *Schistosoma haematobium* SchHae_2.0 [104] and *Schistosoma bovis* ASM395894v1 [105] were downloaded from the WormBase ParaSite (https://parasite. wormbase.org/index.html). An un-rooted phylogenic tree was constructed in MEGA 7 on the basis of multiple alignment of full-sequences from *S. japonicum*, *S. mansoni*, *S. haematobium* and *S. bovis*. Tandem duplicated cathepsin B-like cysteine proteases of *S. japonicum* and *S. mansoni* were indicated by bold black lines.
(TIF)

**S6 Fig.** Interactions between the five types of detected AS genes in the (A) egg stage, (B) miracidium stage, and (C) sporocyst stage were visualized using an UpSet plot.
(TIF)

**S7 Fig.** Venn diagram of the overlap of the DE and DAS genes between the (A) miracidium and sporocyst stages, and (B) sporocyst and cercaria stages.
(TIF)

**S1 Table. Summary of sequence statistics for *S. japonicum* RNA-seq data.**
(XLSX)

**S2 Table. Illumina RNA-Seq *S. japonicum* transcript TPM (Transcripts Per Kilobase million) values.**
(CSV)

**S3 Table. List of stage-specific genes (SSG) or stage-enriched genes (SEG) in the four *S. japonicum* larval stages.**
(XLSX)

**S4 Table. Enriched GO terms of SSG and SEG in the four *S. japonicum* larval stages.** Gene ratio is the percentage of total SEG or SSG in the given GO term.
(XLSX)

**S5 Table. Differentially expressed genes (DEG) between the adjacent life stages.**
(XLSX)

**S6 Table. Enriched GO terms of genes differentially expressed between the adjacent life stages.**
(XLSX)

**S7 Table. Enriched GO terms of genes in the eight clusters.**
(XLSX)

**S8 Table. 18 homologs of stem cell markers between *S. japonicum* and *S. mansoni*.**
(CSV)

**S1 Dataset. The HMMER output file and multiple sequence alignment results of peptidase C1 (papain) in the four *Schistosoma* species, and the GFF file of peptidase C1 (papain) genes in the *Sj*V3 genome.**
(ZIP)

**S2 Dataset. AS events in each life stage and the differential AS events between the adjacent life stages.**
(ZIP)

## Author Contributions

**Conceptualization:** Bingkuan Zhu, Wei Hu.

**Data curation:** Shaoyun Cheng, Jipeng Wang, Yan Lu.

**Formal analysis:** Bingkuan Zhu, Yanmin You, Jipeng Wang, Wei Hu.

**Funding acquisition:** Wei Hu.

**Investigation:** Shaoyun Cheng, Bingkuan Zhu, Fang Luo, Xiying Lin, Yanmin You.

**Methodology:** Bingkuan Zhu, Fang Luo, Xiying Lin, Chengsong Sun, Cun Yi, Jipeng Wang, Yan Lu.

**Project administration:** Bingkuan Zhu.

**Resources:** Bingkuan Zhu, Bin Xu.

**Software:** Shaoyun Cheng.

**Supervision:** Yan Lu, Wei Hu.

**Validation:** Bingkuan Zhu, Wei Hu.

**Visualization:** Yan Lu.

**Writing – original draft:** Shaoyun Cheng, Bingkuan Zhu.

**Writing – review & editing:** Shaoyun Cheng, Bingkuan Zhu, Jipeng Wang, Yan Lu, Wei Hu.

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
