## [Decision Letter · Decision Letter 0]

10 May 2021

Dear Professor Hu,

Thank you very much for submitting your manuscript "Comparative transcriptome profiles of Schistosoma japonicum larval stages: implications for parasite biology and host invasion" for consideration at PLOS Neglected Tropical Diseases. As with all papers reviewed by the journal, your manuscript was reviewed by members of the editorial board and by several independent reviewers. In light of the reviews (below this email), we would like to invite the resubmission of a significantly-revised version that takes into account the reviewers' comments. 

The reviewers have done a thorough job of evaluating the paper. The analyses is very standard stuff, and the biological conclusions rather speculative, so the main merit of this work is the data: its a a bit of a missed opportunity here to put these japonicum results in context with previous work, to the extent that this doesn't really move the field forwards terribly far. Nonetheless, I agree with the basic sentiment of all three reviewers: the paper is based on sound analysis of some important data, and will make a valuable contribution to the literature in this area. I also agree with the caveats pointed out by the reviewers, and think the paper needs some work before it is acceptable. Most of the points of concern are mentioned in the review, but I would like to highlight four major issues. The first three in particular I think are essential in getting the paper to the required standard.

1. It is essential that the version of the S. japonicum genome used here must be made available before acceptance.- it isn't enough that 'It is worth noting that our paper of SjV3 is under review now.': it is currently not possible to replicate any of the results here. Ideally, the genome assembly and annotation would be submitted and available from standard INSDC repositories, but it should at least be available by FTP or similar. In a similar point, it is not always clear what genome version or data is being used for particular analyses, and that needs addressing.

2. More details needed of some methods, in particular, as reviewer 2 mentions, exactly what the biological replicates are - e.g. are they from independent infections / animals or aliquots of a single pool of material. This reviewer also mentions that the PCA and hierarchical clustering analysis needs describing properly: both some information on methods, but in particular, how was 8 clusters decided upon (line 369). Nothing is said about how the peptidase C1 genes of Schistosoma haematobium and Schistosoma bovis were identified: are they from the genome annotations? If so, which versions? If not, how were they identified.

3. I agree with reviewer 3 that the peptidase duplication section doesn't feel well-integrated with the rest of the material here. It also certainly cannot be evaluated without being able to understand (and access) the precise assembly and annotation version used. This section is only related to the gene expression data as 'Besides, all the duplicated genes showed 440 the highest expression at the cercaria stage' on line 439-440, and no data is shown to support this. So something needs to be added to help us understand where these paralogs are expressed and how much - why do the authors feel that this material belongs in an RNA-seq paper?

4. As reviewer 3 points out, the paper would be improved by better cross-referencing other transcriptomic work on schistosomes, in particular the more extensive data available for S. mansoni, which now includes several single-cell transcriptomic datasets. I appreciate this kind of scholarly work is time-consuming, but I think at least something needs to be said in the discussion about the extent to which these findings are japonicum-specific and novel. Without it, this seems like a missed opportunity, but also it doesn't really 'do' very much as a paper except present some slightly speculative interpretations of pretty standard analyses.. its mostly about the data resource.

I also append some detailed notes on grammatical and other minor textual issues with the paper to the end of this message to help with revising the paper, if you choose to do so.

We cannot make any decision about publication until we have seen the revised manuscript and your response to the reviewers' comments. Your revised manuscript is also likely to be sent to reviewers for further evaluation.

Sincerely,

James Cotton

Associate Editor

Poppy Lamberton

Deputy Editor

Reviewer's Responses to Questions

**Key Review Criteria Required for Acceptance?**

**Methods**

-Are the objectives of the study clearly articulated with a clear testable hypothesis stated?

-Is the study design appropriate to address the stated objectives?

-Is the population clearly described and appropriate for the hypothesis being tested?

-Is the sample size sufficient to ensure adequate power to address the hypothesis being tested?

-Were correct statistical analysis used to support conclusions?

-Are there concerns about ethical or regulatory requirements being met?

Reviewer #1: -Are the objectives of the study clearly articulated with a clear testable hypothesis stated?

Yes.

-Is the study design appropriate to address the stated objectives?

Yes.

-Is the population clearly described and appropriate for the hypothesis being tested? Is the sample size sufficient to ensure adequate power to address the hypothesis being tested?

The authors do state that 3 biological repeats were done in the Results/Discussion but key details as to how the study was performed are missing – see the comments below.

-Were correct statistical analysis used to support conclusions?

Yes

-Are there concerns about ethical or regulatory requirements being met?

No. 

Generally, the Materials and Methods section requires more detail for the reader to fully understand how the work was undertaken and ensure studies could be replicated. Authors should clearly state the number of biological replicates performed and numbers of animals (rabbits) used. Specific information the authors should provide further details of are listed below. 

Could the authors include a comment on how the cooling of cercariae prior to centrifugation may impact the transcriptome of this life cycle stage?

• Egg isolation and hatching methodology – lines 134-37.

• How were the repeat experiments performed? e.g. were there independent infections/hatchings? 

• The amounts/volumes of parasite material used for RNA extraction.

• Provide details of the input parameters used for each bioinformatic analysis. E.g. the read trimming criteria used with the fastp tool. 

• Provide references for all bioinformatic tools used.

• Authors write that the genome edition used here is currently under review elsewhere, once published they should update this manuscript with the appropriate reference.

• I couldn’t find a clear description of the methods/tools/parameters for clustering genes by expression -for data shown in Figure 2D.

• For the Peptidase C1 identification and phylogenetic analysis section: authors only mention the 2-species analysis, not the 4-species included in the supplementary figures.

Reviewer #2: Appropriate methods used, all very clearly described.

Reviewer #3: -Are the objectives of the study clearly articulated with a clear testable hypothesis stated?

Yes-ish. Given that it is a largely observational study, authors have a clear rationale for their work. In the introduction they don’t lay out any new/unanswered questions that they are trying to address, for example things that remained unclear after previous similar studies.

-Is the study design appropriate to address the stated objectives?

Yes, the life stages assessed are appropriate. Harvesting miracidia from “artificial pond water” (whatever that is?) sounds fairly reasonable, dissecting sporocysts from snail sounds a bit labour intensive but appropriate, and cercariae harvesting followed by rapid cooling sounds like a good plan. In RNA-Seq it is always very tricky to harvest the right tissue in the right stage, in as “normal” setting as possible. For example, larvae exposed to tap water might have very different expression profile to larvae exposed to actual pond water. 

-Is the population clearly described and appropriate for the hypothesis being tested?

The description is quite clear, with some exceptions like “artificial pond water”. The samples are not quantified, so we don’t know if the RNA contains 1,10 or 100 individuals.

-Is the sample size sufficient to ensure adequate power to address the hypothesis being tested?

The have 3 replicates for each of four life stages, which is usually deemed adequate for this type of study. Read count and mapping counts seem to be in normal range.

-Were correct statistical analysis used to support conclusions?

Yes, their bioinformatics pipeline was very standard. Suitable corrections for multiple p-vals etc. Some very downstream stats were a bit home-made, but in a way that made sense.

-Are there concerns about ethical or regulatory requirements being met?

No

**Results**

-Does the analysis presented match the analysis plan?

-Are the results clearly and completely presented?

-Are the figures (Tables, Images) of sufficient quality for clarity?

Reviewer #1: -Does the analysis presented match the analysis plan? Are the results clearly and completely presented?

Generally, I found the discussion of results a little too brief/vague. I feel the work would be improved by a more detailed discussion in places. Examples:

• ‘Gene expression information correlated well with the well-described biology of each life stage’ (Line 114). Here a brief summary of what aspects of the biology are relevant/interesting here would help the reader. Were there changes the authors expected/predicted to see? The same applies to Line 243.

• GO enrichment analysis was used to interrogate the differences between samples. I think the analysis would be improved by also looking for Pfam enrichment (or similar) as GO coverage is variable in parasite genomes. 

• Reported putative stage-specific leishmaniolysin expression (line 347). Are there sequence/functional/localisation differences between these groups of genes that might offer a hypothesis for this?

• Lines 363-365 – vague.

• Gene clustering based on transcript abundance (lines 368-392 and figure 2D) – the clusters for groups 3/4 seem quite similar to me in the figure. Can the authors discuss why they are separate clusters?

• Lines 385-386 – vague.

• It would be good to include the HMMER output statistics for the peptidase C1 (including p-values) and sequence alignments – likely as a supplementary figure. That would help the authors address the following:

o How similar are the genomic/predicted protein sequences for the tandemly duplicated C1 peptidases? In relation to this, can the authors comment on the mapping statistics for reads over the tandemly duplicated genes? Are the reads uniquely mapped to a single copy?

• The alternative splicing (AS) analysis raw data tables using to make figure 4A should be included in the supplementary, including p-values and gene IDs of alternatively spliced genes. This analysis is simply stated with little discussion. One gene is highlighted without justification in Figure 4F. Authors briefly compare the types of AS in larval stages and adults (line 502), more information to summarise the findings in adult stages would be useful to the reader here. 

Some smaller recommendations/comments: 

• Line 73 – The statement that this work provides functional information for S. japonicum is slightly overstating results. I would suggest rewording this. 

• Lines 252-259 –A direct link between Omega-1 function and the differentially abundance putative T2 ribonucleases is unclear. As such I think authors need to be more cautious in the wording here. If there are additional references/evidence supporting these speculations, they should be included. E.g. Is there evidence for function validation/co-localisation/co-expression of these putative ribonucleases with Omega-1? 

• Line 289 - reference 40 appears to be a transcript focused paper and does not verify correlation of protein and transcript abundances here, consider rewording. 

• Line 317 – it is not clear to be that the statement ‘CaBP may be essential for the infectious cercaria’ is well justified. 

• Line 329 - The reason for the use of a foldchange cut off > 5 should be stated.

• Line 351 – ‘These data indicate that leishmanolysin and neprilysin maybe involved in the penetration of the snail host by the miracidium.’ – not well justified, consider rewording.

• The use of subheadings might help the flow of the ‘Stage-specific genes (SSG) and stage-enriched genes (SEG)’ section.

• Where specific genes are mentioned the gene IDs should be put in the text too. E.g. Lines 299-300 and 269 etc.

• Line 254 – unexplained acronym ‘SEA’. 

• Line 307 – This GO term appears to be wrong. I think it should be 5509.

• Authors use ‘associated with the transition from’ a few times (e.g. line 325,342) I think this isn’t strictly accurate. Consider rewording. 

• The authors could take greater care with some of the wording in their Results/Discussion regarding use of ‘expression’. The work describes the larval stages transcriptomes and transcript abundances do not always correspond to protein abundance. It is suggested that the authors use more conservative terms like ‘transcript abundance’ rather than ‘gene expression’.

-Are the figures (Tables, Images) of sufficient quality for clarity?

Generally, the figure legends are too brief, many are missing information important to interpret the figures (see below for details). 

I would suggest moving the supplementary figures 3A, C, E to the main figures as these plots nicely describe the data. Similarly, I would recommend moving 4B to the supplementary figures. (Not required, merely suggestion)

Figure 1A – ‘PCA results’ is not descriptive enough here.

Figure 1C – clarification of ‘differentially expressed’ and ‘commonly’ expressed as this was not immediately clear.

Figure 2A – On figure, ‘Specifical’ change to specific?

Figure 2B – I found this figure difficult to interpret with the GO accession numbers on the plot, I would recommend putting these with the text to the left. What is mean by the ‘Gene Ratio’ should be stated in the legend, additionally in the legend for Table S2.

Figure 2D – ‘Membership’ metric was not well explained. State y-axis units.

Figure 3B – Missing units on x-axis bar. Clustering on right of figure squashed making it difficult to see clusters – this may benefit from labels indicated the discussed clusters. The colouring of motifs/genome features clashes a bit, can you make them more distinct?

Supplementary Figures 1 and 2 – label the legend in the figure to clearly indicate what score the colour gradient is reflecting. More detail is needed as to how these scores are arrived at, either in the figure legend or in the Materials and Methods section. 

Supplementary Figure 3

• For A/C/E - some points are cut off from the plot, the y-axis should be adjusted. Include what the dotted lines represent in the legend. Additionally, the axis scale appears to be Log2 foldchange but the highlighted datapoints are based on fold change – this was a little confusing. Check point highlighting for p-values, some non-significant points appear to be coloured. 

• B/D/F – the legends are quite small making them difficult to read. GO term numbers would be useful to add to these also.

• 3A – Can the authors comment on the skew to down negative fold changes show in the plot? 

• 3B – there is no discussion of the magnesium ion binding GO enrichment in Miracidium the main text. Not essential but consider discussing.

Figure 4F – including the structure (exons, introns etc.) of the whole transcript or genomic sequence at the base of this plot would aid interpretation of the isoforms shown.

Figure S4 – are all duplication events indicated on this tree? I think the S. mansoni ones from the main figure are missing. It would also be good to add in any from the two additional species. 

Finally, I would encourage the authors to consider including a graphical abstract/figure highlighting the main findings in a schematic of the life cycle stages. This would help communicate the key findings and summarise the work.

Reviewer #2: Excellent and thorough presentation of results

Reviewer #3: -Does the analysis presented match the analysis plan?

Yes, it does. All results reported have clear methods sections which explain how they got there, and their choice of analysis methods are overall appropriate. The one bit that is missing is a method/cut-off for alternative splicing and alternative transcript reconstruction. I just cannot follow how they go from mapped reads to predicting alternatively spliced gene models, to determining if there is significantly different transcript expression patterns between life stages.

-Are the results clearly and completely presented?

Yes, results are clearly presented. 

-Are the figures (Tables, Images) of sufficient quality for clarity?

The PCA plot looks good, and supports good data quality – although the biological replicates look very very similar to each other.The choice of figures, and the supplementary tables are excellent. It might have been nice to have a table of all read counts/FPKMs? But apart from that all you expect is there.

**Conclusions**

-Are the conclusions supported by the data presented?

-Are the limitations of analysis clearly described?

-Do the authors discuss how these data can be helpful to advance our understanding of the topic under study?

-Is public health relevance addressed?

Reviewer #1: -Are the conclusions supported by the data presented?

Overall, the conclusions discussed are supported by the data presented.

-Are the limitations of analysis clearly described?

I could not find a clear discussion of the limitations of the study – this would be useful to address comments relating to the methods section e.g. cooling of cercariae during harvesting. I would also be interested to read about any ideas the authors had for priorities and next steps from this work in the concluding remarks.

-Do the authors discuss how these data can be helpful to advance our understanding of the topic under study?

Yes. 

-Is public health relevance addressed?

Briefly, this could be expanded.

Reviewer #2: Conclusions are appropriate and supported by data.

Reviewer #3: -Are the conclusions supported by the data presented?

Some of the conclusions feel a bit speculative, eg. 261-275 the authors suggest that CFAP could either be “involved in the assembly of the cilium” or involved in chemical detection of the snail host, based on CFAP role in mice. Those are both fine hypotheses, which could be tested, but seems a bit speculative to present as a result for this present study. In contrast, the discussion 270-275 about 5-HT contains references to much more relevant and specific studies.

The results 277-296 also contains relevant references, and suitable comparison to similar species and similar results, however, it only shows that what is seen in S. japonicum is exactly what is expected, and has been known for >10 years. 

The very nice section about tandem genomic events in C1 (394 ff) is interesting, but the main take-home is the argument that several cathepsins have undergone genomic duplication. And since this paper is not publishing a new S. japonicum genome, or even making one publicly available, it is very hard to validate this section. S. japonicum genomes have also been published before, and it would be nice to be offered a view into if their findings can also be seen in the prjna520774 genome assembly. Maybe this whole section would fit better into the genomic paper, because the final take-home message relating to expression data in fig 3d is not novel, but merely a confirmation what is already known. Not to nit-pick, but the HMMer analysis of PF00112 has already been pre-calculated for you in Wormbase ParaSite if you go to the BioMart tool (which shows 10 C1-containing proteins, rather than the 29 or 30 reported by the authors – and the lower number is more in line with what is expected from S.mansoni. Of course we cannot know for sure which is right, and it is possible that the current S. japonicum genome assembly and annotation is lacking, and it would be worth trying to tease it out which is more right. But this paper does not convince me that they have done all the job needed to prove it either way. I do look forward to the genome publication coming out, to see if this question has been addressed more in-depth there – which would in my opinion be the more appropriate place for that whole section.

-Are the limitations of analysis clearly described?

Fairly well. The parts which are more speculative are recognisable as such. The writing is lacking a bit of the “flair”, and preciseness which comes from more experience and technical skill in writing a scientific paper. It is fully functional and readable – but a bit more technical writing skill might have avoided some of the minor corrections I suggested, if the sentence had been more skillfully crafted and elegantly written.

-Do the authors discuss how these data can be helpful to advance our understanding of the topic under study?

Not really. They do a fairly good job of summarising the overall output of their analysis, but they do no experimental validation of any of their hypotheses regarding gene function. Although these are important parasites, the analysis does not really go beyond just describing some of their findings, and occasionally putting it into context of what we already know. There is not really any novelty there. No experimental validation, and not proper comparison with previous studies on the topic.

-Is public health relevance addressed?

No. This paper is a nice confirmation of some things we already knew – which in a way validates that their data and analysis is okay – but it does not give any new insights of public health relevance.

**Editorial and Data Presentation Modifications?**

Reviewer #1: I have included comments relevant to this section in the ‘Results’ section as I felt they were clearer there.

Reviewer #2: N/A

Reviewer #3: Minor: In the methods section, expand a little bit on how many individuals were combined into each sample? Also – how were the biological replicates made? Eg parasites extracted once, and split into three samples, or taken 3 times on different days?

Minor: Throughout the text there are a few instances of abbreviations being introduced without being spelled out, eg line 254 “SEA”, which authors might want to clarify in this context means “soluble egg antigen”. “C 1” that is referred to multiple times in the text might be known to more readers as “Cysteine peptidases” or more specifically cathepsin, which is of large importance due to their role in schistosoma vaccine research eg PMC3897446, and the relevance of those were not introduced in the introduction, although they feature in the discussion. Minimising abbreviations makes the text more readable.

Minor: Paragraph 252-259 is a bit hard to follow.

Minor: I’m not sure I agree with the statement that “AS has been investigated in S. japonicum schistosomula and adult worms [69,70], but these analyses were based on the highly fragmented V1 version genome (SjV1) that will cause the loss of information.” The PRJEA34885 genome is quite fragmented, but prjna520774 is comparatively not too bad compared to other Schistosoma genome assemblies. And the “fragmentation” in genomes more often occur in highly repetitive gene-poor regions, so a fair few alternative splicing events can be correctly inferred even in a highly fragmented genome. 

Major: Alternative splicing patters in prjna520774 are available for anyone to browse at https://parasite.wormbase.org/jbrowse/index.html?data=%2Fjbrowse-data%2Fschistosoma_japonicum_prjna520774%2Fdata&loc=scaffold_84%3A3696001..4170000&tracks=DNA%2Ccmscan_rfam_12.2.nucleotide_match%2Crepeatmasker.repeat_region%2CGene_Models%2Cdust.low_complexity_region%2Ctrf.tandem_repeat&highlight= so what the authors have done in terms of predicting alternative splicing is not particularly new. The also have not shared the genome or gene prediction they have done, and not validated its accuracy to current versions of genes and gene models. The instinct that the authors have in going to look for alternative splicing patterns between the different life stages is great, and potentially could come up with some interesting patterns. Figure 4d is really quite neat though, I like it more and more the more I look at it.

Minor: 500-502 reference missing for the statement “previously detected in schistosomula”.

Minor: 503-506 The statement “SE is thought to be the most prevalent AS type in animals, whereas IR represents the most common AS form in plants and unicellular eukaryotes [72]. Hence, our results suggest that the gene regulation pattern of S. japonicum is much closer to its mammalian hosts.” I don’t think anyone who has ever seen the S. japonicum would disagree with the authors, but this is also clearly not a unique and new finding, or even unexpected, since S. japonicum are animals, and not unicellular eukaryotes.

Major: They just have not put their results properly into the context of the other studies of differential expression of Schistosoma egg, miracidium, sporocyst and cercaria. It would not be that hard to download and re-analyse previously published data, and more carefully go through similarities and differences. Most of their main results are either things that are already known, or relates to an unpublished genome.

Minor: The data accessions cited are not yet publicly available: PRJNA719283, SRR14133806-SRR14133817. (assuming they will be)

**Summary and General Comments**

Reviewer #1: The authors describe and compare the transcriptomes of the S. japonicum larval stages using RNAseq. This will be a welcome resource for the community. The authors go on to identify and characterise duplication events in the peptidase C1 family (members of which have important role infection) as well as examining their transcript abundances in each larval stage. Finally, the group begin to evaluate the alternative splicing across the transcriptomes. 

Whilst the analysis is good greater detail is needed in both description of the methods and in the discussion of results. The importance of some results are not well stated. Additionally, the introduction is a little brief and could describe in more detail some of the key morphological and molecular differences between larval stages to aid the non-expert reader in interpretation of their work. Whilst the work is understandable, I think it would benefit from additional copy editing for grammar/sentence structure – though this is absolutely not a major concern.

Reviewer #2: This is a very thoroughly analysed and nicely presented piece of work, which gives new insight into the intramolluscan transcriptome landscape of Schistosoma japonicum. I recommend publication subject to corrections of minor issues.

My only criticism is whether the coverage of analysis is a little selective - this is always the case with papers like this, authors naturally focus on what is interesting to them, and you cannot please everybody. But I was just left wondering about expression of for example neoblast (stem cell) related genes, which are en vogue in the helminth literature recently, could the authors consider examining these to see if they inform when and where stem cell replication is occurring in the life cycle?

Line 108: Authors should note here that a transcriptome is already available for intramolluscan S. mansoni (doi: 10.1371/journal.pntd.0007013), but since that one was generated from whole snails, it highlights the novelty of the current approach in which separate ex vivo life stages have been analysed.

Line 146: Indicate volume/mass of parasite tissue used for RNA extraction from each life stage. You specify three biological replicates per library in results, I suggest mentioning that here also.

Line 176: correct "manosoni"

Reviewer #3: In this paper, authors have done RNA-Seq of egg, miracidium, sporocyst and cercaria of S. japonicum – a zoonotic parasitic worm. It is an achievement, because these are very small, it is difficult to extract and enrich them fast enough, and in enough quantity to yield good sequencing. The authors have done 3 biological replicates of each life-stage and RNA-Seq, which has been thoroughly analysed using standard and appropriate methods. The paper aims to give new insights into an hitherto underexplored part of the S. japonicum life cycle, but I’m not sure it quite does. They should have mentioned that the same and more life stage expression profiling has been done previously by Gobert 2009 PMC2670322 (not referenced) and Cai 2017 PMC5223471 (referenced). Very similar studies have also been conducted for closely related Schistosoma species, and in my opinion this paper would be enriched for more carefully comparing and contrasting their results with those of previous studies. They do find their results much overlap with what already is known, which is showing that the study was probably technically quite well executed. But going into a bit more detail comparing/contrasting with other would have given them more to talk about. Or alternatively – pick the most interesting story and going in for more proper experimental validation of the different ideas raised?

PLOS authors have the option to publish the peer review history of their article (what does this mean?). If published, this will include your full peer review and any attached files.

Reviewer #1: Yes: Megan Sloan

Reviewer #2: No

Reviewer #3: No

Figure Files:

Data Requirements:

Reproducibility:

*** 

detailed (minor) comments from the AE:

1. citations and software version numbers are missing for some software packages: DeSeq, STAR, RSEM, tximport, MEME. 

2. as reviewer 3 requests, a reference to the 'artifical pond water' (or composition) would be helpful on line 136.

3. As the reviewers note, there are some problems with the writing: mostly the usual errors of singular vs plural and articles that many non-native English writers make, and are easily fixed. I've tried to list as many of these as possible below, together with some other minor corrections. In the below '->' means 'should be replaced by'.

line 37: miracidium : should read miracidia

lines 37"We also found that miracidium might use leishmanolysin and neprilysin to penetrate the snail, while elastase (SjCE2b) and leishmanolysin might contribute to the host invasion by cercaria." seems like a possible overinterpretation of their results, and known?

line 85: excluded -> excreted

line 88: penetrates snail host assisted with secretions -> 'penetrates the snail host assisted by secretions'.

line 90: loses the ciliated plates -> 'loses its ciliated plates'.

line 91:  After the asexual multiplication -> After a period of asexual multiplication

line 94: Once contact with skin -> On contact with the skin

line 107: gene expressions -> gene expression

line 114: 'We identified stage specifically or enriched expressed genes that could be vital for the dominant functions of the parasite in specific life stage'  -> 'We identified genes specifically expressed or with enriched expression in each stage that could thus be vital for the dominant function of the parasite in those life stages'.

line 136: the artificial pond water -> artificial pond water

line 139: 'purified after three times of washing' -> 'purified by washing three times'

line 165-166: Was differential expression applied to every gene? Its usual to remove genes with low total numbers of reads across all stages before comparison, to improve power post-multiple correction testing.

line 172 - We need to be told which accessions from NCBI were used here: as far as I can see, Sjv2 is on NCBI, but most of the paper is based on v3 - why the discrepancy?

line 176: manosoni -> mansoni

line 178 : un-rooted -> unrooted

line 180-181: were classified into different groups based on the classification scheme : its not at all claer what is meant by 'the classification scheme' here.

line 187: again, which reference?

line 202: false discovery rate (FDR) < 0.05 This is quite a lenient cut-off for FDR. IT would be interesting to know something about how many of the identified AS events are well-supported at lower FDR.

line 227: because a part of the eggs -> because some eggs 

line 229: inter-sample Venn diagram analysis -> its not clear what is being shown here. Are genes counted as being expressed in a particular stage if there is even a single RNA-seq read? OR is there some cut-off here? There is some definition of stage-specific on line 244: does this apply here too? If so, it seems quite loose. It would be nice to know how this changes if you allow a few reads (e.g. 5 or 10) in other stages. 

line 244: its not clear what 'significantly' means here: presumably some cut-off in false discovery rate was applied?

line 252: Egg induces granuloma in host. -> Eggs induce granulomas in the mammalian host.

line 258: I think some recognition that Ribonuclease T2 proteins have very diverse functions is probably warranted here 

line 261: Miracidium is free-swimming and penetrating snail host -> Miracidia are free-swimming and penetrate the snail host

line 261: 'many members': I think we should be told how many.

line 266: development of miracidium -> development of miracidia

line 267: Since miracidium is attracted -> Since miracidia are attracted

line 275: movement in miracidium -> movement in miracidia

line 277: Sporocyst residents in snail host and produces cercariae -> Sporocyst are resident in snail hosts and produce cercariae

line 286: located in -> present in.

line 290: at the daughter sporocyst -> in the daughter sporocyst

line 298: Cercaria is free-swimming and invading mammalian host -> Cercariae are free-swimming and invade the mammalian host

line 301: release of cercaria -> release of cercariae

lines 306,308, 313, 314, 315. cercaria -> cercariae

line 310: during the development -> during development

line 332: Compared to miracidium, fifteen GO categories were upregulated in egg, -> Compared to miracidia, fifteen GO categories were upregulated in eggs

line 343: upregulated in miracidium -> upregulated in miracidia

line 348: expressed in sporocyst -> expressed in sporocysts

line 350: This claim appears to go far beyond the evidence presented.

line 356: of sporocyst -> of sporocysts.

line 370: the sentence starting: 'Cluster 1, 2, 3' doesn't really make sense. I think it needs to read. 

clusters 1,2 and 5 showed highest expression in eggs, miracidia and cercariae respectively, while clusters 3 and 4 were both most highly expressed in sporocysts'.

line 379: These GPCR -> these GPCRs

line 495: I think the result referred to here is the kind of comparisgon shown on the Venn diagram (fig 4e) - comparing differential gene expression with differential alternate splicing.. the text is ambiguous, however, and seems to suggest the comparisons are with differential gene expression and the total amount of alternate splicing in a stage. This needs clarifying.

line 527: the skin penetration -> skin penetration

lines 528-529: might also involve in -> might also be involved in

line 542: I don't think an acknowledgement section needs to be included if there are none!

line 571, 610-611, 638-639, 644, 656, 675, 713: its not usual to list the editor of a journal article in the reference listl

---

## [Editor Report · Decision Letter 1]

14 Sep 2021

Dear Professor Hu,

Thank you very much for submitting your manuscript "Comparative transcriptome profiles of Schistosoma japonicum larval stages: implications for parasite biology and host invasion" for consideration at PLOS Neglected Tropical Diseases. As with all papers reviewed by the journal, your manuscript was reviewed by members of the editorial board and by several independent reviewers. The reviewers appreciated the attention to an important topic. Based on the reviews, we are likely to accept this manuscript for publication, providing that you modify the manuscript according to the review recommendations. 

I think the authors for their careful and thorough revision of the manuscript. I think it is very nearly ready for acceptance. There are some relatively minor problems with the grammar of many of the new text sections in the paper. These are listed below. Please note line numbers are from the 'changes highlighted' word document version of the manuscript, not the revised PDF.

The most important problem is that the reference genome is still not available, which makes this work impossible to replicate in its current state. The authors now provide an accession number for a BioProject, but that is currently not visible - I guess under embargo pending a genome paper becoming available.. but in the context of the current manuscript, we have no guarantee of the availability of these data on any defined timeframe. I think the authors need to either make the bioproject publicly available before this paper can be accepted, or make these data available for review somewhere (e.g. FTP or, preferably, an alternative data repository). 

There are two new sections of text (added in response to reviewer comments) that attempt to explain how the highest-level RNA-seq differences observed correlate with the expected biological differences between different developmental stages of schistosomes, but they are so high-level as to not really do anything useful, and rather repetitive between the two. There is already a much better description of the life-cycle in the Introduction. I would just delete these two bits, and leave in a sentence like: (e.g. at line 394): "Since the parasite at each stage shows distinct biological characteristics, we attempted to identify genes expressed specifically at each stage, or significantly more highly expressed at each stage. We thus defined..". At line 147 I would delete the first sentence 'Miracidium embryo was developing inside the immature egg', as it doesn't really tell us anything about the pattern of gene expression expected.

*****starting line 147*** : Miracidium embryo was developing inside the immature egg. Miracidia and cercariae showed high motor and proteolysis activity, ready for the host invasion. DNA replication and cell division only occurred in the sporocysts. 

***line 394: ***Since the parasite at each stage shows distinct biological characteristics, e.g., the embryonic larva (miracidium) is developing inside the immature egg; the miracidia and cercariae are highly mobile and store many proteases for the host penetration; germinal cells exist in the sporocysts, as asexual polyembryonic process is taking place in this stage.

line 40-42: The expression profile of the stem cell markers revealed the potential germinal cell conversion during the larval development  The expression profile of stem cell markers revealed potential germinal cell conversion during larval development

lines 53-54: of the S. japonicum larval stages and identifies a set of genes that might involve in the intermediate and definitive host invasion.  of S. japonicum larval stages and identifies a set of genes that might be involved in intermediate and definitive host invasion.

lines 77-78: Additionally, it indicated that different populations of germinal cells maybe existed in the larval stages. ->

Additionally, it indicated that different populations of germinal cells may exist in the larval stages.

line 159 - maybe -> 'may be'

line 186 - to be absolutely clear, I think the sentence about rabbit infections could do with an 'each' at the end.

line 196, were served as a biological replicate -> were used as a biological replicate

lines 208, 220: were served as one replicate. -> were used as one replicate 

line 212- after three times of washing -> after washing three times 

line 246: After the clusters generation -> After cluster generation

line 261: The Principal Component Analysis -> Principal Components Analysis

line 415: T2 ribonuclease was one of the top 25 genes enriched in eggs showed in the previous study - would be better as  T2 ribonuclease was identified as one of the top 25 most highly enriched genes in eggs in a previous study.

line 652: The dot line  The dotted line

line 712 : study the schistosome stem cells, -> study schistosome stem cells,

line 725: Then we described the expression files of -> Then we described the expression profiles of

line 738: relative high expression -> relatively high expression

line 929: schistosomula -> schistosomulae

line 932, 954: IR  RI

line 986: the penetration of the miracidia to the snail. -> the penetration of the snail by the miracidia.

***line 952-954: Could differences in AS event type frequency be due to technical differences between the different experiments, rather than species differences? 

line 992 - 994: Expression profile of the stem cell markers revealed that different populations of germinal cells were existed in the larval stages. -> The expression profile of stem cell markers revealed that different populations of germinal cells exist in larval stages.

Sincerely,

James Cotton

Associate Editor

Poppy Lamberton

Deputy Editor

Please note line numbers are from the 'changes highlighted' version of the manuscript.

I think the authors for their careful and thorough revision of the manuscript. I think it is very nearly ready for acceptance. There are some relatively minor problems with the grammar of many of the new text in the paper. these are listed below. Please note line numbers are from the 'changes highlighted' word document version of the manuscript, not the revised PDF.

The most important problem is that the reference genome is still not available, which makes this work impossible to replicate in its current state. The authors now provide an accession number for a BioProject, but that is currently not visible - I guess under embargo pending a genome paper becoming available.. but in the context of the current manuscript, we have no guarantee of the availability of these data on any defined timeframe. I think the authors need to either make the bioproject publicly available before this paper can be accepted, or make these data available for review somewhere (e.g. FTP or, preferably, an alternative data repository). 

There are two new sections of text (added in response to reviewer comments) that attempt to explain how the highest-level RNA-seq differences observed correlate with the expected biological differences between different developmental stages of schistosomes, but they are so high-level as to not really do anything useful, and rather repetitive between the two. There is already a much better description of the life-cycle in the Introduction. I would just delete these two bits, and leave in a sentence like: (e.g. at line 394): "Since the parasite at each stage shows distinct biological characteristics, we attempted to identify genes expressed specifically at each stage, or significantly more highly expressed at each stage. We thus defined..". At line 147 I would delete the first sentence 'Miracidium embryo was developing inside the immature egg', as it doesn't really tell us anything about the pattern of gene expression expected.

*****starting line 147*** : Miracidium embryo was developing inside the immature egg. Miracidia and cercariae showed high motor and proteolysis activity, ready for the host invasion. DNA replication and cell division only occurred in the sporocysts. 

***line 394: ***Since the parasite at each stage shows distinct biological characteristics, e.g., the embryonic larva (miracidium) is developing inside the immature egg; the miracidia and cercariae are highly mobile and store many proteases for the host penetration; germinal cells exist in the sporocysts, as asexual polyembryonic process is taking place in this stage.

line 40-42: The expression profile of the stem cell markers revealed the potential germinal cell conversion during the larval development  The expression profile of stem cell markers revealed potential germinal cell conversion during larval development

lines 53-54: of the S. japonicum larval stages and identifies a set of genes that might involve in the intermediate and definitive host invasion.  of S. japonicum larval stages and identifies a set of genes that might be involved in intermediate and definitive host invasion.

lines 77-78: Additionally, it indicated that different populations of germinal cells maybe existed in the larval stages. ->

Additionally, it indicated that different populations of germinal cells may exist in the larval stages.

line 159 - maybe -> 'may be'

line 186 - to be absolutely clear, I think the sentence about rabbit infections could do with an 'each' at the end.

line 196, were served as a biological replicate -> were used as a biological replicate

lines 208, 220: were served as one replicate. -> were used as one replicate 

line 212- after three times of washing -> after washing three times 

line 246: After the clusters generation -> After cluster generation

line 261: The Principal Component Analysis -> Principal Components Analysis

line 415: T2 ribonuclease was one of the top 25 genes enriched in eggs showed in the previous study - would be better as  T2 ribonuclease was identified as one of the top 25 most highly enriched genes in eggs in a previous study.

line 652: The dot line  The dotted line

line 712 : study the schistosome stem cells, -> study schistosome stem cells,

line 725: Then we described the expression files of -> Then we described the expression profiles of

line 738: relative high expression -> relatively high expression

line 929: schistosomula -> schistosomulae

line 932, 954: IR  RI

line 986: the penetration of the miracidia to the snail. -> the penetration of the snail by the miracidia.

***line 952-954: Could differences in AS event type frequency be due to technical differences between the different experiments, rather than species differences? 

line 992 - 994: Expression profile of the stem cell markers revealed that different populations of germinal cells were existed in the larval stages. -> The expression profile of stem cell markers revealed that different populations of germinal cells exist in larval stages.

Figure Files:

Data Requirements:

Reproducibility:

References

---

## [Editor Report · Decision Letter 2]

8 Oct 2021

Dear Professor Hu,

We are pleased to inform you that your manuscript 'Comparative transcriptome profiles of Schistosoma japonicum larval stages: implications for parasite biology and host invasion' has been provisionally accepted for publication in PLOS Neglected Tropical Diseases.

Best regards,

James Cotton

Associate Editor

Poppy Lamberton

Deputy Editor

---

## [Editor Report · Acceptance letter]

6 Jan 2022

Dear Professor Hu,

We are delighted to inform you that your manuscript, "Comparative transcriptome profiles of Schistosoma japonicum larval stages: implications for parasite biology and host invasion," has been formally accepted for publication in PLOS Neglected Tropical Diseases.

Best regards,

Shaden Kamhawi

co-Editor-in-Chief

Paul Brindley

co-Editor-in-Chief
